# Distributed acoustic sensing of microseismic sources and wave propagation in glaciated terrain

F. Walter [1]✉, D. Gräff [1], F. Lindner[1], P. Paitz[2], M. Köpfli[1], M. Chmiel[1] & A. Fichtner[2]

Records of Alpine microseismicity are a powerful tool to study landscape-shaping processes and warn against hazardous mass movements. Unfortunately, seismic sensor coverage in Alpine regions is typically insufficient. Here we show that distributed acoustic sensing (DAS) bridges critical observational gaps of seismogenic processes in Alpine terrain. Dynamic strain measurements in a 1 km long fiber optic cable on a glacier surface produce high-quality seismograms related to glacier flow and nearby rock falls. The nearly 500 cable channels precisely locate a series of glacier stick-slip events (within 20–40 m) and reveal seismic phases from which thickness and material properties of the glacier and its bed can be derived. As seismic measurements can be acquired with fiber optic cables that are easy to transport, install and couple to the ground, our study demonstrates the potential of DAS technology for seismic monitoring of glacier dynamics and natural hazards.

[1] Laboratory of Hydraulics, Hydrology and Glaciology (VAW), ETH Zürich, Switzerland. [2] Institute for Geophysics, ETH Zürich, Switzerland.
✉email: walter@vaw.baug.ethz.ch

Over the past 1–2 decades, advances in sensor and digitizer technologies have increased portability of seismic instrumentation. Seismic monitoring in poorly accessible Alpine and Polar regions is therefore becoming increasingly feasible. The resulting data focus on processes near the Earth's surface rather than on traditional seismology subjects like the deeper crust and mantle.

Seismic studies in Alpine terrain have cultivated new sub-disciplines like environmental seismology[1] and cryoseismology[2,3]. This has filled critical observational gaps for investigation of mass movements such as bedload transport in torrents[4], rockfalls[5], debris flows[6], and avalanches[7] as well as the stability of rock structures[8] and landslides[9]. In glaciated regions, seismic studies have proved the existence of seismogenic glacier sliding and provided time series of iceberg production[2] and subglacial water flow[10], which are difficult to obtain with traditional glaciological measurements.

The sub-second time scales at which seismometers monitor ground unrest constitute an unrivaled temporal resolution. However, only specialized and dense sensor networks can locate mass movements with reasonable uncertainty[5]. Capturing, for instance, precursory signals before failure requires sensors in the immediate vicinity of the unstable mass[9], and coincidental recordings from nearby permanent seismic stations are rare and difficult to interpret[11]. Therefore, comprehensive, large-scale seismic monitoring essential for early warning or scientific purposes remains largely impossible because the required sensor coverage is usually infeasible.

In other fields of seismology, the advent of distributed acoustic sensing (DAS) is currently revolutionizing seismic sensor coverage. DAS technology uses fiber-optic cables into which an interrogator injects a sequence of laser pulses. The time series of back-scattered signals can be transformed into strain rate sampled every few meters along the fiber[12]. Seismic waves dynamically straining the fiber can thus be recorded over distances of several tens of kilometers, with a bandwidth ranging between quasi-static and tens of kHz[13]. Advanced interferometric techniques applied to ultra-stable laser light injected into fiber-optic cables may even sample cables hundreds of kilometers long[14].

Fiber-optic technology such as DAS has started to complement geophone chain deployments in active exploration surveys (e.g., ref. [15]) and has been shown to record waveforms of regional and teleseismic earthquakes[16,17]. In sea basins, seismograms measured with fiber-optic cables capture local earthquake signals, which are too weak to be recorded by sparse ocean bottom seismometers[14]. DAS measurements of anthropogenic noise can furthermore be used to characterize the Earth's near-surface structure[18,19]. Though individual DAS channels may have a lower signal-to-noise ratio (SNR) than conventional seismometers, the presence of unused fibers in telecommunication networks (dark fibers) suggests that a vast resource of already installed seismic sensors could be harnessed for monitoring with an unprecedented sensor coverage and density[16,19].

Here we present DAS measurements of microseismic signals and ambient noise records acquired on a Swiss Alpine glacier. For 5 days in March 2019 we monitored glacier stick-slip activity, rockfalls, and crevasse icequakes using a fiber-optic cable placed on the glacier surface. The cable layout formed an equilateral triangle with 220-m long sides. Compared with the records of 3 collocated on-ice and 3 nearby on-rock seismometers, utilizing the nearly 500 DAS channels provides a significant improvement in stick-slip event location and identifies previously unnoticed critically refracted and multiply reflected seismic waves. The DAS measurements furthermore recover the back azimuth of a visually confirmed rockfall and yield subsurface velocity estimates from passive noise correlations. These results show the utility and potential of DAS measurements for monitoring glacial processes and mass movements in high Alpine regions.

## Results

**Rhonegletscher.** The study site is located on the ablation zone of Rhonegletscher (Switzerland), a temperate glacier (ice at pressure melting point) with an area of ~15.5 km² and a length of ~8 km flowing southward from 3600 to 2200 m above sea level (a.s.l.) at an average surface slope of 10° (Fig. 1[20]). Between 20 and 26 March 2019, we carried out DAS measurements close to the glacier's central flow line at an altitude of ~2500 m a.s.l. (network center coordinates: 2,672,300, 1,161,050 (LV95), 46.5968, 8.382 (WGS84)). At our study site, the surface velocity of Rhonegletscher is ~35 m/a

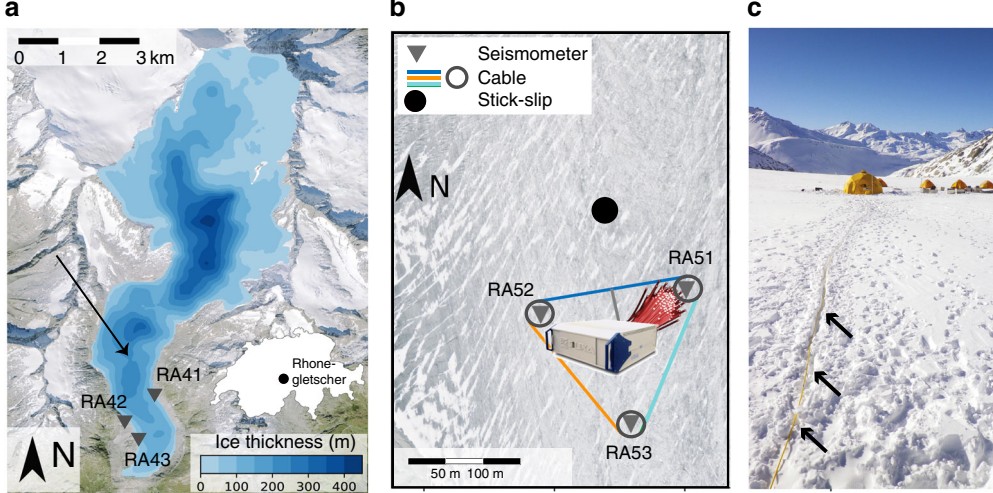

**Fig. 1 Study site on Rhonegletscher. a** Glacier thickness from interpolated radar profiles[21] on orthophoto of year 2014 (ice flow from North to South). Triangles indicate on-rock seismometers (model LE3D 5s) and black arrow points toward the location of the region shown in panel (**b**). **b** Network layout of seismometers and fiber-optic cable as well as location of stick-slip event shown in Figs. 2, 3, and 4, and located with the DAS records. Orthophoto shows site at almost snow-free conditions in summer whereas the DAS measurements for the present study were conducted on a 3 m snow cover. Residual snow bridges show local crevasses. **c** Photo of fiber-optic cable in snow (arrows) and field camp (photo by Manuela Köpfli). Orthophoto was provided by Swisstopo: ©/with permission/2020 swisstopo (JD100042).

and the ice thickness reaches 200 m as determined from inter-polation of radar transects[21] and hot-water drilling in summer 2018[22]. Throughout the 1-week-long DAS field deployment the glacier ice was covered by ~3 m of snow.

**Instrumentation**. Three on-ice seismometers were deployed close to the central flow line of the glacier and form an equilateral triangle with ~220 m side lengths (Fig. 1b). Each consists of three-component Lennartz 3D/BHs sensors, drilled ~3 m into the ice, and a Centaur digitizer by Nanometrics. The sensor's eigen-frequency is 1 Hz; the response is flat up to 100 Hz. We sample these sensors at 500 Hz. Three nearby on-rock stations were deployed on granite bedrock within few tens of meters of the glacier margin (Fig. 1a). They consist of three-component Lennartz 3D/5 s surface sensors with 0.2 Hz eigenfrequency and a flat response up to 50 Hz, and a Centaur digitizer sampling at 200 Hz. In the past the glacier bed beneath the on-ice seismometers had produced repeated microseismic activity. We therefore monitored this region for nearly 2 years and chose it as the field site for this present study.

From 21–25 March 2019 we deployed the SILIXA iDAS™ fiber otpic system at Rhonegletscher. The interrogator was placed into a tent, powered by a Honda 20i generator (2 kW maximum power output) and connected to a 1-km long polyurethane mantled fiber-optic cable containing four single-mode and two multimode fibers. We utilized two single-mode fibers to increase spatial redundancy by splicing the end of one fiber together with the other fiber. This resulted in a total fiber length of 2 km. Approximately 820 m of the cable were placed into a shallow trench (few cm deep) carved into the snow and subsequently covered with loose snow (Fig. 1c). Of these, 660 m form an equilateral triangle with additional cable segments shaped into loops of ca. 10-m diameter around the triangle corners, defined by the locations of the on-ice seismometers (Fig. 1b). A ca. 30 m cable segment connected the northern triangle side to the interrogator. The triangular layout was chosen to facilitate comparison between seismometer and DAS records in this particular glacier region.

For the temporal and spatial sampling of the DAS deployment we chose 500 Hz and 4 m, respectively, during most of the measurement period. This configuration was briefly changed to 4000 Hz and 8 m when explosives at ca. 30-cm depth in the ice were set off within 10 m of the northern triangle side. The goal of these explosions was to evaluate the DAS system's performance in active seismic experiments on glacier ice. For the entire measurement period, we used a gauge length of 10 m, which is the distance over which the interrogator calculates dynamic strain rates[13]. This gauge length is smaller than the seismic wavelengths of primary interest and thus does not alter seismic arrival-time measurements (Supplementary Notes 1–4). We determined the DAS channel locations with differential GPS and foot taps to within an uncertainty of 2 m, i.e., half the spatial sampling distance.

**Recorded signals**. On Alpine glaciers, dominant seismic signals from surface crevasse activity, englacial water flow and nearby rockfalls range between a few Hz and tens of Hz[2,10]. Basal seismicity recorded at the surface has frequencies that may exceed hundreds of Hz[23].

Figure 2 and Supplementary Figures 1–4 show our DAS records of a surface icequake, a more impulsive stick-slip event, an explosive charge, and a sustained 15-s-long signal of a rockfall. The rockfall was visually observed in the field and thus associated with the sustained signal (Fig. 2d). The impulsive event (Fig. 2b) is identified as a basal stick-slip event based on waveform characteristics, location, and higher frequency content compared with the surface icequake (Supplementary Figs. 1 and 2; further

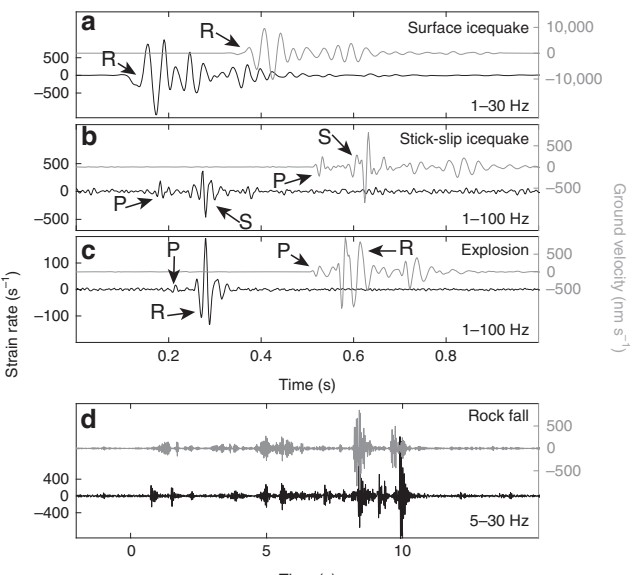

**Fig. 2 Microseismic events. a** Bandpass filtered seismograms (vertical seismometer record in gray and DAS record in black) of surface icequake, **b** stick-slip icequake, **c** explosion, and **d** rockfall. Time series in panels (**a–c**) were recorded at southern triangle corner (seismometer RA53 and DAS channel D620). Time series in (**d**) was recorded at western triangle corner (RA52 and channel D904). Note that the time axes between DAS and seismometers records in panels (**a–c**) were slightly shifted for illustration purposes. Filter corners are specified. For surface icequake (**a**), stick-slip icequake (**b**), and explosion (**c**), P-, S- and/or Rayleigh phases are indicated.

explanations follow). It belongs to a cluster of stick-slip events repeating every few hours and producing highly similar wave-forms (Fig. 3a). Except for bandpass filtering and amplitude scaling we show unprocessed time series to compare the signal quality between DAS and the vertical component of the seismometer (Fig. 2). Assuming, for simplicity, that the incoming wave is nearly planar, strain measured on the fiber is proportional to particle velocity, which makes the recordings qualitatively comparable[15]. An accurate comparison requires rotating the horizontal seismometer records along the cable axis. Since we use borehole sensors, the needed rotation angle is not a priori known.

The surface icequake shows the characteristic dominant Rayleigh wave between 10 and 50 Hz (Supplementary Fig. 1) with a retrograde elliptical particle motion[24]. In contrast to the surface icequake, the stick-slip event has a higher frequency content and it shows dominant P and S arrivals while lacking a notable Rayleigh phase. However, even on the DAS system, which in principle is sensitive in the kHz range[13], the frequency content fades out at frequencies above 100–200 Hz (Supplementary Fig. 2). We explain this high-frequency limit primarily by differences in coupling between seismometers and the fiber-optic cable: The seismometers were drilled into the ice and tightly frozen into their boreholes, which provides an ideal coupling to the ice. On the other hand, the cable rests on over 2 m of damping snow, which provides a poor coupling to the ice body. Explosions in glacier ice are known to contain energy up to 1000 Hz[25] but in our case the snow damping suppresses frequencies above 100–200 Hz (Supplementary Fig. 4) on the DAS system compared with the seismometer records (Fig. 2c, Supplementary Figs. 4 and 5). Snow damping also explains why high-frequency reflections from the explosions are visible on the seismometers but not on the DAS record (Supplementary Fig. 5). As expected for a near-surface source, the explosion seismogram also shows the dominant Rayleigh phase.

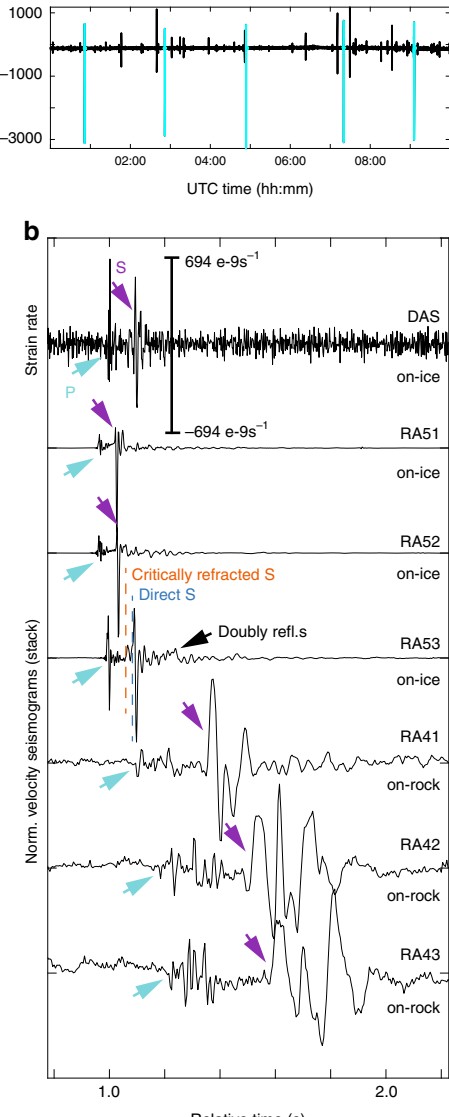

**Fig. 3 Stick-slip occurrence and waveforms. a** Ten hours long continuous, unfiltered record of on-ice station RA52. Cyan time series portions are events belonging to the stick-slip multiplet. **b** Stick-slip seismogram recorded with DAS channel D620 (at southern corner) and stack of 43 on-ice and on-rock vertical seismometer records. Arrivals of P- and S-waves are indicated. For the on-ice stations, the P-arrival is dominated by the direct wave, although a faster wave traveling partially within the underlying bedrock may induce a minor precursory signal. The direct P-wave motion is compressive (upward) in agreement with the double-couple mechanism representing a shear dislocation whose bed-parallel hanging wall slips along the general ice flow direction. At on-ice station RA53, direct and indirect wave phases separate and can be identified with help of the analysis shown in Fig. 4. For the on-rock seismometers, the first arrival cannot be explained with a direct P-wave arrival but instead is a critically refracted P-wave traveling through the bedrock. The polarity of the refracted P-wave is dilatational (down) as this wave samples another quadrant of the double-couple radiation pattern than the direct wave[26].

Figure 2 shows that in the frequency range between several Hz and 100 Hz, the SNR of the DAS records is below the on-ice seismometers. This has been observed in other contexts[16]. Nevertheless, the DAS system provides clear records of surface icequakes, stick-slip events, rockfalls, and other strong signals not

shown here (e.g., helicopter and sustained harmonic wave trains of anthropogenic origin).

The signal strength and SNR of DAS records vary spatially as shown for the case of the stick-slip event in Supplementary Fig. 6. As expected, signal strength and SNR tends to be strongest on the northern cable segment, which is closest to the source. There exist additional variations of signal and noise strength. In particular, channels along the eastern portion of the northern cable segment (between channels D1768 and D276) tend to have lower SNR. Snow depth variations measured with an avalanche probe along the northern cable section amount to ca. 60 cm, which seems minor compared with the systematic SNR variation along these channels. Instead, we find it more likely that snow quality differences (wet vs dry snow) and varying contact areas between snow surface and cable explain variations in signal and/or noise strength.

**Event location and stick-slip magnitude**. Figure 4b (cyan and black point clouds) shows the probabilistic location inversion of the stick-slip event shown in Figs. 2 and 3 using on-ice seismometer and DAS arrival times with the density of scatter points representing the probability density of the hypocenter location (see Methods). As a result of uncertainties in the seismic velocity model and arrival-time picks and the nonlinear inversion problem, the probability density using the three on-ice seismometers (cyan point cloud in Fig. 4b) has large side lobes including local minima. The 1 sigma uncertainty ellipse has semimajor and minor axes of 142 and 107 m. In contrast, for the arrival times measured with the more numerous and spatially denser DAS channels, the probability density function is substantially more confined (black point cloud in Fig. 4b) with semimajor and minor axes of 35 and 11 m.

Given the known source location and typical properties of glacier ice, the time integral of the horizontally polarized S-wave recorded on the DAS system can be used to estimate stick-slip moment magnitude[26]. For this we furthermore assume a source mechanism consistent with bed-parallel slip along the glacier flow line, which agrees with compressive first motions on all on-ice stations (Fig. 3). The estimated moment magnitude lies in the range of −1.5 to −0.5, depending on the exact fault plane orientation, which is poorly constrained with the given seismic data. This estimate is comparable to other accounts of microseismic basal stick-slip icequakes[26].

**Phase identification of stick-slip icequake**. A cross-correlation search matching the stick-slip seismogram in Fig. 3 as a template against the continuous seismometer record shows that the event belongs to a multiplet of repeated ruptures over identical fault planes resulting in practically identical seismic waveforms (Fig. 3a). During the DAS deployment, 48 repeating events matched the template with correlation coefficients of on-ice seismometer records between 0.986 and 0.999. The inter-event times are remarkably regular at around 2 h. Compressive P-polarities are consistent with a shear dislocation along the glacier flow direction (Fig. 3b).

A record section of the stick-slip icequake on the DAS system highlights P- and S-waves and additional phases (Fig. 4a). 2D ray tracing (see Methods) suggests that the recorded P-wave train contains both the direct wave as well as the critically refracted phase, which travels through the underlying granite (the cross over distance at which the refracted wave passes the direct wave is similar to the smallest source-station offset). The same is true for the S-wave, but here a separation between direct and refracted phase is clearly visible at source-station offsets beyond 270 m (Fig. 4d). A small arrival (visible at distances above 310 m) before the refracted S-wave may be explained with a doubly reflected

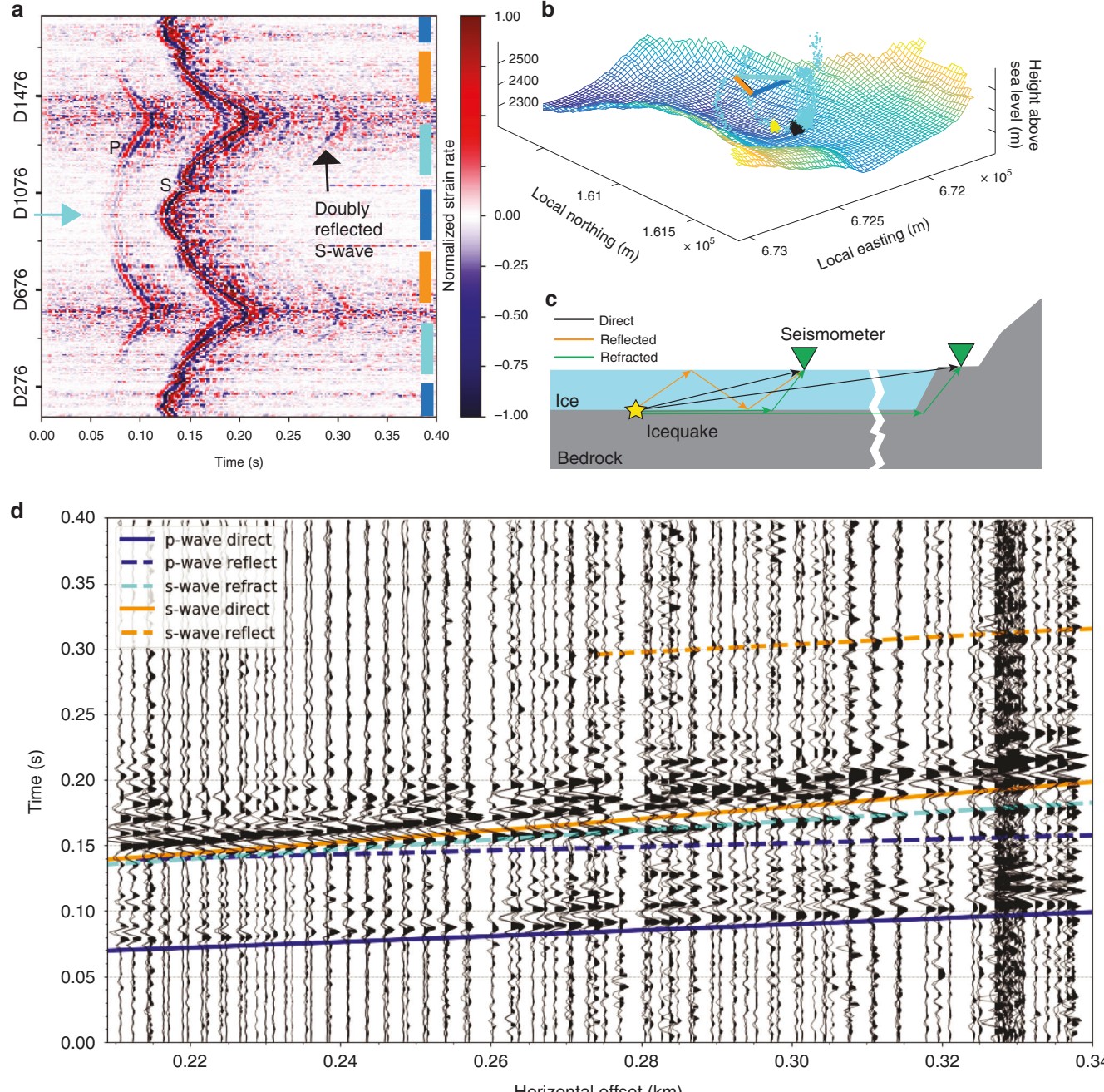

**Fig. 4 Record sections and probabilistic source location of stick-slip event. a** Full DAS record of normalized stick-slip seismogram. The DAS laser is reflected at the cable end near Channel D988 (cyan arrow). Higher channels sample similar cable locations to the ones below Channel D988 giving similar (though not identical) strain rate seismograms causing a symmetrical appearance with respect to Channel D988. The bar colors on the panel right correspond to the cable segment colors of panel (**b**) and Fig. 1b, indicating channel locations (space in between the colored bars corresponds to cable loops). P- and S arrivals are indicated. Differences in relative P and S amplitudes result from the double-couple radiation pattern of the stick-slip event and different angles between wave polarization and cable axis. **b** Bedrock topography (color corresponds to elevation) and location of stick-slip event: black and cyan dots, respectively, indicate location grid search using arrival times of the DAS and seismometer records of the event shown in panel (**a**). Yellow point cloud is the location of the event shown in Supplementary Fig. 11. Point density is proportional to location probability density. The seismometer arrival times give rise to side lobes and larger location uncertainties than the DAS arrival times. Triangle represents cable layout with colors corresponding to bars on the right of panel (**a**). **c** Schematic of waves traveling between stick-slip icequake and recording stations. **d** Record sections and theoretical arrival-time estimates using an adjusted 2D velocity model of ice over bedrock (see Methods for details).

P-wave (Fig. 4c). The latest indicated arrival (best illustrated in Fig. 4a at around 0.3 s) points toward a doubly reflected shear wave arrival, although the calculations place the arrivals slightly after the signal onset, which likely results from our simplified 2D velocity model.

The refracted and doubly reflected S-wave arrivals are also visible in on-ice seismometer records (Fig. 3b). However, without the dense sensor layout of the DAS cable, these phases are more difficult to interpret. On the DAS system, Fig. 4d shows additional coherent arrivals after the direct S-arrival. These may be other

multiple reflections involving conversion between P- and S-polarization at the surface or bed.

The bedrock S-velocity tuned with the 2D ray tracing is 3200 m/s. The ice thickness needed to match the arrival times is 151–181 m. This is smaller than the 172–202 m estimated on the basis of interpolated radar lines (Supplementary Fig. 7) and several boreholes drilled in summer 2018 near the study site showing a depth between 187 and 200 m. However, the interpolated radar lines spaced by hundreds of meters have considerable uncertainty and do not capture details in bed topography, which gives rise to a bed slope of up to 25° and a bed overdeepening beneath the study site (Fig. 1 and Supplementary Fig. 7). Given that we neglect 3D ice surface and bed topography in our simple 2D ray-tracing model, our ice thickness estimates seem reasonable. In general, however, the source-station geometry exhibits an azimuthal gap of more than 270°, which is too large for precise joint inversion of hypocenter and seismic velocities (ref. [27] and see Methods).

In contrast to the on-ice records, the first arrivals recorded on the rock stations travel at up to 4600 m/s (neglecting topography) and thus cannot be explained with a direct phase propagating through the ice. We therefore interpret this first arrival as the critically refracted P-wave. The polarity of this refracted P-wave is opposite of the direct P-arrival of the on-ice stations (Fig. 3b), because the direct and critically refracted waves sample different quadrants of the double-couple radiation pattern of basal stick-slip events[26].

**Rock-fall back azimuth**. The rockfall (Fig. 2d) induces a coherent signal throughout the fiber-optic cable. Small time shifts of rock-fall seismograms between individual channels are a result of different arrival times. Previous studies have exploited such signals on seismic arrays to locate rockfalls and image their trajectories[28]. Here, we use matched-field processing to determine the rockfall's back azimuth and the apparent velocity at which the seismic waves propagate throughout the cable layout based on signal arrival-time differences (see Methods).

Figure 5 shows the back-azimuth and velocity calculation for a time window containing the rockfall signal. For the times before and after the rock-fall signal, the normalized beam power is consistently below 0.2 indicating poor signal coherence throughout the array (see Methods). During the rockfall signal, coherence nearly doubles and back azimuths are stable at 243° East from North, pointing toward an unstable moraine, ~1 km to the West of the glacier where the rockfall was visually observed (Fig. 5b). Moreover, phase velocities of 1700 m/s (Fig. 5c) agree with typical Rayleigh wave velocities below 30 Hz of crevassed near-surface ice[29], which is consistent with superficial rock impacts on the ground. Besides the unstable moraine from where the rockfall in Fig. 5 detached, matched-field processing shows activity on slopes east or south of the array, although these sources were not visually confirmed.

**Noise correlations**. In the 5–50 Hz range, cross-correlations of the continuous DAS record from 24 March 2019 can be used to estimate the phase of the fundamental-mode Rayleigh wave (Fig. 6; see Methods section). The cross-correlation wave packets propagate along the eastern side of the triangle. The acausal part of the noise correlation is largely absent, because ambient noise sources are not homogenously spread around the study site[30]. The propagation of coherent noise signals is mostly (though not entirely, Fig. 6b) in the southwestern direction with noise sources locating to the northeast of the study site. As a consequence, noise correlations using the northern triangle side include less coherent noise propagating along the cable axis and therefore have a lower SNR (not shown). With an absence of melt-water flow during the

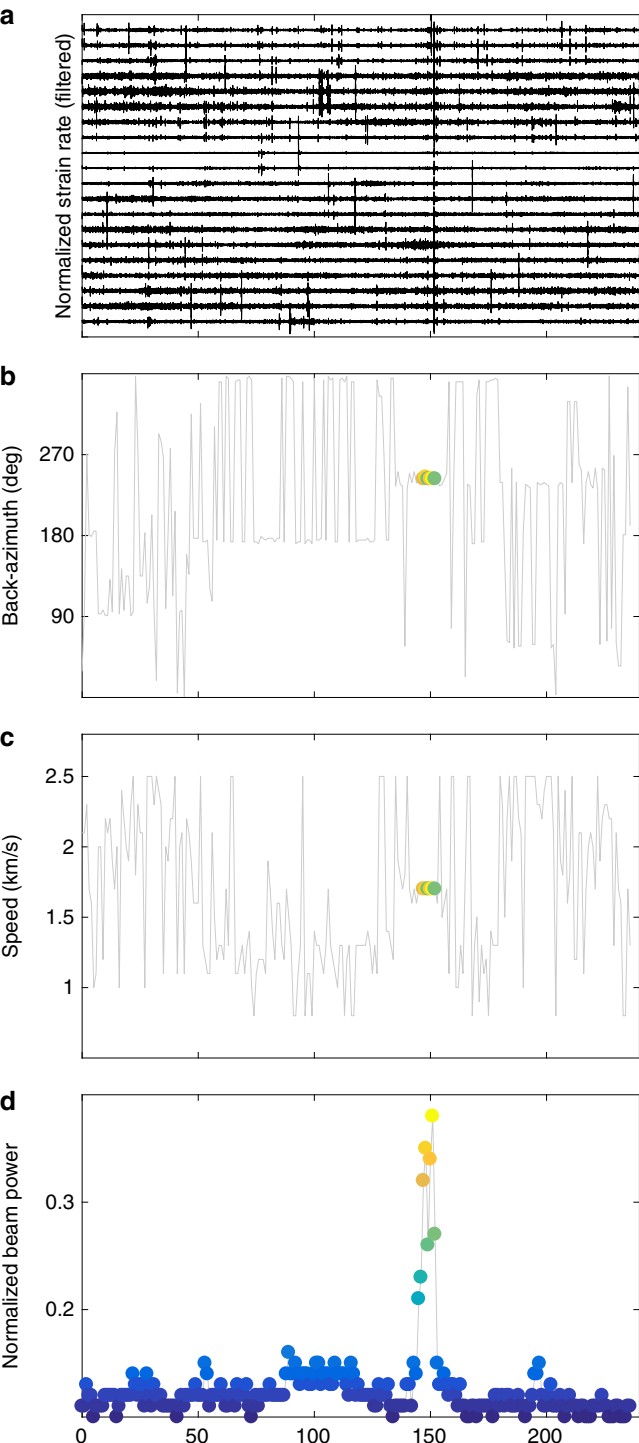

**Fig. 5 Matched-field processing (MFP) of a DAS rock-fall seismogram.**
**a** 20 DAS channels filtered between 10 and 30 Hz. Notice the frequent occurrence of noise transients at amplitudes comparable to the rock-fall signal around 150 s. **b** Calculated back azimuth. **c** Calculated phase speed. **d** Normalized beam power showing a sudden increase of signal coherence within the DAS channels when the rock-fall signal is recorded. Color code represents normalized beam power and is the same as in panels (**b**) and (**c**). This shows that during the rockfall, the calculated back azimuths and phase velocities are stable, but jump between the grid search extremes at other times. During the rock-fall signal, back azimuth and phase velocity are consistent with Rayleigh waves emitted by visually observed rock impacts on the ground west of the instrumented site.

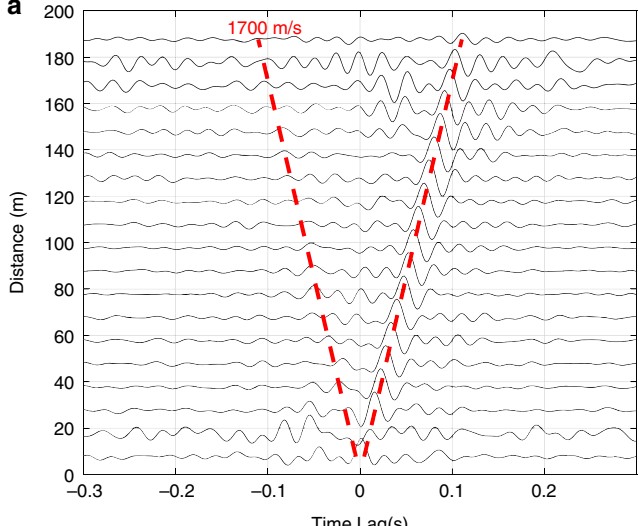

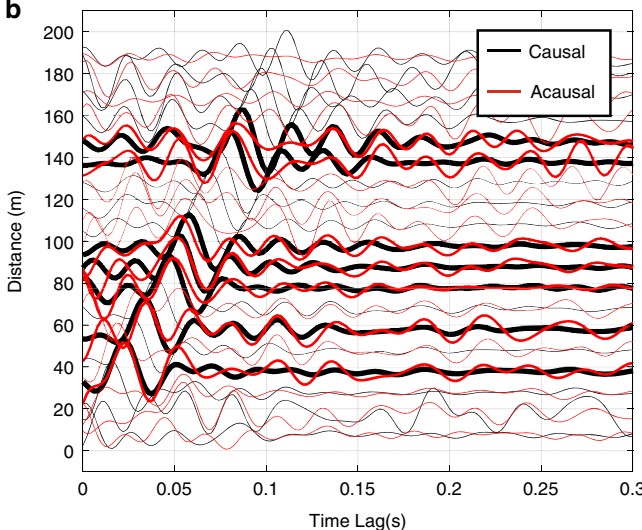

**Fig. 6 Noise correlation stacks.** Recorded on 24 March 2019, the stack includes 1081 cross-correlations from 47 channels of the eastern triangle side binned within fixed distance intervals of 10 m. **a** Cross-correlations showing Rayleigh wave propagation at 1700 m/s. **b** Acausal part folded onto the causal part of the cross-correlation. Although the acausal SNR of the virtual surface wave is weaker than the causal one, comparison of the two nevertheless shows that some energy propagates in both directions along the cable. Bold traces show the highest symmetry between the causal and acausal waves.

measurement, which would facilitate noise interferometry[31], crevasse activity to the northeast of the study site likely provides coherent signals in the background seismicity.

The virtual Rayleigh waves propagate at a typical[29] velocity of 1700 m/s (Fig. 6a). The DAS channels record strain in the direction of the fiber, induced by the elliptical particle motion along the Rayleigh wave propagation axis. In principle, the dispersion relation of virtual Rayleigh waves could be used to infer ice thicknesses. However, in our case, the 220 m length of straight cable portions inhibits resolution below 10 Hz (Supplementary Fig. 8), where Rayleigh waves are sensitive to the glacier bed[29,31].

## Discussion

Similar to applications in other seismological disciplines, DAS technology offers a vast potential for monitoring glacier dynamics and Alpine mass movements. With a simple deployment procedure essentially consisting of rolling out a cable, hundreds of seismic measuring points are available for monitoring. The physical labor is comparable to installation of only a few seismometers at the ice surface which produce significantly less information about seismic sources and wave propagation within and near the glacier.

Our results show that the DAS system is capable of recording seismogenic glacier flow and even small Alpine mass movements such as rockfalls. Compared with seismometers on and near the glacier, the DAS system offers clear advantages, which allow us to better constrain static and dynamic properties of the glacier and its surroundings. A first important result is that the DAS records allow for accurate arrival time measurements despite the spatial averaging of dynamic measurements implied by the finite gauge length (Supplementary Notes 1-4 and Supplementary Fig. 9). Consequently, even though the DAS cable covered the same area as the 3 on-ice seismometers, the amount and density of recording channels improved the location quality of stick-slip events substantially. Furthermore, the close spacing of recording DAS channels reveal the existence of multiple reflections and critically refracted waves. These phases cannot be identified with sparsely spaced seismometer networks and had previously only been observed beneath the polar ice sheets[26,32] or not at all. Finally, the application of matched field processing to DAS data allows us to locate rockfalls, and thus, to identify potentially unstable slopes.

The advantages of the DAS system outweigh the lower SNR of individual channels along the fiber-optic cable compared with our borehole seismometers in direct contact with the glacier ice. At frequencies above 100–200 Hz, part of the low SNR can be attributed to the highly damping 2–3 m snow cover separating the fiber-optic cable from the ice surface. Placing the cable directly on the glacier ice, e.g., before winter snow fall would likely increase the quality of seismic records and mitigate variations in signal and noise amplitudes along the cable, which we attribute to snow quality and coupling variations.

In the present study, identification of the indirect phases emitted by the stick-slip source would not have been possible without the DAS system. In our case, the resulting estimates of ice thickness and seismic velocities of the glacier bed substrate were subject to uncertainties resulting from a poor azimuthal sensor coverage of the source hypocenter. However, with longer cable segments and better sensor coverage, our measurements offer new perspectives in cryoseismology. Without the need of active sources, DAS measurement can characterize the subglacial environment. In our case, seismic velocities within the glacier bed are higher than ice as expected for a mountain glacier resting on granite bedrock. This is confirmed by proglacial terrain, which until recently was covered by the tongue of the Rhonegletscher (Supplementary Fig. 10). In contrast, for the largest tide water glaciers and ice streams on Earth, whose dynamics control eustatic sea-level rise[33], weak basal till layers allow for rapid basal motion of up to tens of meters per day and till layers of 100 s of meters thickness are thus characteristic for fast ice-stream flow (e.g., ref.[34]). Especially when water saturated, such till layers have low seismic velocities compared with ice and bedrock[35]. Stick-slip seismicity could thus provide important information about the basal boundary conditions of fast polar ice streams. The fact that stick-slip patches tend to produce repetitive events such as shown here and in previous studies[23,26] furthermore suggests an application for monitoring: Small changes in basal seismic velocities revealed by repeating stick-slip events could help identify changes of basal resistance as a result of evolving subglacial water pressures (e.g., ref.[36]).

In our deployment it was sufficient to place the DAS cable into a cm-deep snow trench. Two persons were enough to deploy

hundreds of meters of cable within a few hours, resulting in 500 recording channels. Covering the cable layout with geophones or seismometers at equivalent sensor spacing instead would have required significantly more manpower and time. The straight-forward cable deployment implies that larger areas of a glacier can now be covered with seismic sensors. Covering the full extent of Rhonegletscher with a flow-line-parallel cable of around 10 km therefore seems realistic. With such a layout, a key question concerning ice flow could be answered: do microseismic stick-slip events affect overall ice flow? As a result of technical limitations, this question has been addressed only with few seismic networks monitoring limited regions of glaciers and ice streams[23,37]. With accumulating seismic evidence for seismogenic stick-slip motion[2,3], we have yet to understand the role of these events in ice flow and clarify if conventional theories of glacier sliding, which neglect friction[38], have to be revised. DAS measurements monitoring a full glacier extent could finally test the hypothesis if basal slipperiness determined from numerical models[39] is related to stick-slip activity.

Large-scale DAS measurements on glaciers would not only provide important information on basal seismicity and englacial fracturing. A longer cable would decrease the low corner frequency of ambient noise interferometry. As a result, surface wave phases in noise correlations would be sensitive to the glacier bed, thereby providing ice thickness estimates without the need for active sources. In general, we expect a significant increase in seismic signal quality when placing the DAS cable on snow-free ice surfaces during summer conditions when absorption of short wave radiation tends to heat up cables and melt them into the ice. We also expect a better SNR of noise correlations in the presence of surface melt[31].

With recording channels spaced every few meters along a fiber-optic cable, large data volumes result and efficient data analysis becomes a challenge. In microseismic studies, machine learning algorithms have proven useful for detecting near-surface seismic sources[40] and identifying noise time series suitable for inter-ferometric studies[19]. These approaches could be applied to DAS records and enhanced with array techniques[41] such as matched-field processing used here. Moreover, records from geophones or seismometers installed sparsely along the fiber-optic cable can help to efficiently scan DAS records for repeating glacier stick-slip events: template searches such as shown in Fig. 3 can precisely determine detection times of stick-slip repeaters on seismometer or geophone records. These detection times can subsequently be used for a DAS signal stack, whose SNR is expected to increase with the square root of the number of stacked repeater signals. Even for five events belonging to a cluster producing relatively weak stick-slip events (hypocenters shown as yellow point cloud in Fig. 4b), this stacking substantially improves the SNR and brings out phases, which are not visible for DAS records of individual events (Supplementary Fig. 11). In essence, this approach leverages both the higher SNR from seismometers or geophones for event detection and the DAS system's dense sensor coverage.

In addition to glacier-related seismic records, the DAS system also recorded typical signals of Alpine mass movements, one of which was a visually confirmed rockfall (Fig. 5). The rockfall involved only a few individual blocks with a total volume amounting to a few cubic meters or less. Despite this small size, the DAS system recorded a clear signal, and further processing provides a well-constrained and stable back azimuth, and thus a location of rock impacts on the ground. With a cable placed directly on the ice surface or buried into the ground, the SNR of such mass movement recordings will increase. At the same time, our study shows that fiber-optic cables with comparatively poor coupling (in our case via a damping snow layer) are nevertheless capable of detecting even small Alpine mass movements over hundreds of meters. This suggests that fiber networks of tele-communication lines can be used for mass movement monitoring. Although such cables were deployed for communication purposes in shafts designed to reduce frictional coupling to the ground, they have been used for detecting earthquakes[14,16,17]. Our results suggest that similar detections could be made for Alpine mass movements. With fiber-optic networks already installed in many Alpine regions and along roads, train lines or other infrastructure, DAS technology could in the near future significantly lower detection thresholds and increase warning capabilities for destructive mass movements.

## Methods

**Arrival-time picking and location.** For the seismometer records of the stick-slip event we picked the first breaks of the P-arrival and the direct S-wave (Supplementary Fig. 9), assigning an uncertainty of one sample (2 ms). The direct S-wave can be distinguished from the S-wave critically refracted within the underlying bedrock as explained below. For the P-wave, the direct wave dominates over the refracted one, but both blend into each other and can hardly be distinguished (Fig. 4).

As the wavelengths in our analyzed seismograms exceed the gauge length of the fiber-optic cable, arrival times can also be accurately picked on the DAS system (Supplementary Notes 1-4 and Supplementary Fig. 9). However, for the DAS record, picking the first breaks of direct waves was less reliable as a result of the much lower SNR. Therefore, we picked the maxima of the direct S-wave from 40 channels that are equally distributed along the triangle sides. We could distinguish the direct S-wave from refracted arrivals with an uncertainty of 1 sample (2 ms). In addition, we also picked as many P-wave arrivals as possible, mainly from the southern cable section. Here, P-waves tend to have higher amplitudes, which can be explained by a combination of enhanced P-radiation in the down-glacier direction (assuming an along-flow slip) and different angles between P-wave polarization and cable axes. The latter effect explains why relative P-wave amplitudes are particularly low near the center of the northern triangle side (cyan arrow in Fig. 4a). Similar to the seismometer records, the refracted and direct P-waves are more closely spaced and thus difficult to distinguish. We picked the maxima of the first arriving phase with an uncertainty of 2 ms. In order to account for our convention of picking maxima rather than first breaks, and for the different frequency contents of P- and S-waves leading to an apparent later arrival of the lower frequency S-wave maxima, we assigned a total uncertainty of 10 ms.

We applied a probabilistic nonlinear hypocenter location scheme (NonLinLoc[42]) that accounts for picking and velocity model uncertainties. Sensor location uncertainties are not accounted for in this method. To overcome this limitation for the DAS channels, we divide the channel location uncertainty of ±2 m by an assumed P- and S-wave velocity ($v_p = (3800 \pm 200)$ m/s, $v_s = (1900 \pm 100)$ m/s; see below), which translates into a location uncertainty of 0.5 ms × $v_p$ and 1 ms × $v_s$. Finally, we add this location uncertainty linearly to the picking uncertainty of 2 ms for the S-wave and 10 ms for the P-wave. In total, the picking uncertainty used for the DAS records is 11 and 3 ms for P- and S-waves, respectively.

In order to locate the impulsive stick-slip event shown in Fig. 2b, we separately inverted phase arrival times with NonLinLoc measured on the three on-ice seismometers and the DAS recordings. For the velocity model, we use $v_p = (3800 \pm 200)$ m/s and $v_s = (1900 \pm 100)$ m/s. These values are slightly higher than what was used in a previous seismic study on an Alpine glacier[23], but agree with the 2D velocity model used for ray tracing (see discussion below). Note also that as a result of the surface crevasse zone with near-vertical fracture orientation, significantly lower seismic velocities have been determined near the surfaces of glaciers[43], including Rhonegletscher[44]. However, we consider this effect negligible for our basal source, whose seismic rays cross significantly less near-surface fractures than seismic rays emitted by shallow sources. Similarly, we neglect the effect of the snow layer on travel times, because travel time uncertainties associated with highly variable seismic velocities in snow[45] are likely comparable to uncertainties associated with our homogeneous velocity model assumption.

We estimate the uncertainty of the body wave velocities to ±5% and use a homogeneous velocity model over the entire domain and refrain from including underlying bedrock as ice thickness is known only approximately below our study site: Ice thickness from radar measurements[21] is spatially interpolated, and an uncertainty of at least 10% of the ice thickness should be assumed. Previous source locations have shown that accurate 3D bedrock topography models allowing for critically refracted waves within the bedrock may slightly improve the hypocenter location accuracy over the homogeneous halfspace of ice assumed here[46].

**2D ray tracing and choice of seismic velocities.** For the 2D ray-tracing model we extract a longitudinal glacier cross section along the axis defined by the south-ernmost corner of the fiber-optic cable and a point within the one sigma location uncertainty of the stick-slip epicenter (Supplementary Fig. 7). Horizontal offset shown in Fig. 4d is measured from this point. We apply a 342-m-wide moving

average filter to the cross-sectional bed profile to suppress bedrock steps, which the ray tracer cannot handle numerically and which are likely spurious features of the radar line interpolation[21]. We also rotate the along-cross-section coordinates 5° counterclockwise to compensate for the glacier surface slope and achieve a flat glacier surface. Although this profile includes the bed overdeepening beneath the study site, it does not capture transverse variations in bed height (up to 60 m) and surface height (up to 4.5 m) at the study site (Supplementary Fig. 7).

We calculate the straight paths of direct waves and use a 2D ray-shooting algorithm[47] to calculate the arrival times of doubly reflected and refracted body waves (Supplementary Fig. 7). P- and S-velocities of 3800 and 1900 m/s are needed to match the steep slopes of the indirect arrivals shown in Fig. 4d, even when the longitudinal cross section is manually thinned by 21 m resulting in a maximal ice thickness of 151–181 m. For the doubly reflected waves, the match is further improved by shifting the reflecting bed region an additional 3 m upward. Such a local thinning is justified by bedrock undulations observed in the glacier forefield. Besides that it may represent reflections that occur off-axis with respect to the longitudinal cross section.

The arrival times from the 2D ray tracing are indicative, only, because of bed topography and location uncertainties, with the latter resulting from poor sensor coverage with an azimuthal gap of more than 270°. A joint inversion of velocity model and hypocenter location would provide better constraints, but this also requires better sensor coverage[27]. For our manual arrival-time fitting, the englacial seismic velocities of 3800 and 1900 m/s for P- and S-waves were chosen to agree with the values used for hypocenter location. These velocities are higher compared with a previous study on Glacier d'Argentière[23], which uses 3600 and 1610 m/s for P- and S-waves. However, S-wave velocities agree to within 2, 4, and 6% of the values of the studies by Deichmann et al.[24], Neave and Savage[48], and Walter et al.[43], and similar or smaller deviations hold for P-waves. Theoretical velocities for isotropic single-crystal ice even slightly exceed our values[49]. Moreover, velocities of up to 2170 m/s have been found for S-waves traveling along the fast direction of fracture-induced anisotropic ice, although these values result from inversion of surface wave dispersion, which is subject to tradeoff with other parameters[29].

Our 2D ray tracing hinges on the match of the doubly reflected S-wave, even though it has a weak amplitude and is only visible on a subset of stations. Abandoning this constraint would allow lower seismic velocities. Our high average velocity of doubly reflected S-waves can be explained if this phase does not only contain direct waves traveling within the ice, but also critically reflected waves: The wave may first travel as a critically refracted phase along the ice-bed interface and then enter the ice medium, upon which it undergoes the two reflections. Since the bed velocity is substantially faster (in our case 3200 m/s), this would decrease the arrival time or lower the S-velocity within ice. The phase moveout shown in Fig. 4 argues for this scenario, because the slope of the doubly reflected wave is more similar to the refracted S-wave than to the direct S-wave. This is best seen in the arrival-time curvatures shown in Fig. 4a. In order to further investigate the possibility of refraction followed by a double reflection, raytracing seems inadequate, because wave amplitudes and phases should be modeled, as well.

Generally, in future studies, longer cable segments could provide better sensor coverage. This would mitigate the location-velocity tradeoff and provide observational constraints for ray tracing or full waveform modeling allowing for 3D variations of bed topography. As a result, bed topography and seismic velocities would be better constrained.

**Matched-field processing (MFP).** MFP exploits signal coherence within the sensor array of the DAS system to calculate source back-azimuth and apparent seismic velocities. Since the DAS records contain noisy channels, we only use channels with SNR exceeding 8.5 (calculated as the ratio between maximum signal amplitude and pre-event noise root-mean-square). For the event shown in Fig. 5, 68 channels fulfill this requirement. We next cut out the rock-fall signal and step through this signal in windows of 2 s with 50% overlap and for each of these windows we apply MFP. For subwindows of 0.2 s (50% overlap), and frequencies $f$ between 10 and 30 Hz (0.2 Hz steps), MFP matches a data vector $d(f)$ against a steering vector $\tilde{d}(f)$. $d(f)$ is an N-dimensional vector whose entries are the sub-window's Discrete Fourier transforms at each of the $N$ sensors (in case of the rockfall shown in Fig. 5, $N = 68$). The steering vector $\tilde{d}(f)$ represents theoretical propagation of a seismic phase in a homogeneous halfspace. The match amounts to an inner product between $d(f)$ and $\tilde{d}(f)$[50] but is formally performed using the cross-spectral density matrix (CSDM), which is defined as the outer product

$$\text{CSDM}(f) = d(f)d^{\dagger}(f) \tag{1}$$

where $\dagger$ is the complex conjugate operation. The inner product between $d(f)$ and $\tilde{d}(f)$ is called the beam power and is a measure for signal coherence and hence the quality of the MFP result. Since we keep only phase information in the CSDM (this amounts to spectral whitening of the signal), we neglect seismic attenuation depending on wave type (e.g., surface vs. body wave) and the beam power is normalized with unity indicating perfect coherence.

**Noise correlations.** We split the continuous DAS record of the eastern triangle side into 30-min-long time windows, which we spectrally whiten to reduce the influence of transient and monochromatic seismic sources. The 47 channels are then cross-correlated to yield 1081 pairs. Stacking all 30-min time windows over a full day (24 March 2019) and channels within a 10-m radius further increases the cross-correlation SNR.

Cross-correlations of strain rate data are a function of the spatial gradients of the inter-station Green's functions and the noise source distribution. Our cross-correlations of axial strain rates along the eastern straight cable portion are most sensitive to Rayleigh wave sources that locate along the cable axis beyond the cable ends[30]. For simplicity, we neglect the influence of the noise source distribution, and assume that the interferometric wavefield is proportional to the empirical Green's function. In this case, the causal and acausal wavelets of the cross-correlation represent Rayleigh waves traveling in opposite directions between the station-pairs, where the weak acausal signal (Fig. 6) is evidence for reduced englacial scattering[50] and noise sources located primarily to the northeast of the study site.

A two-dimensional Fourier transform over time and wavenumber $k$ (defined as $k = 2\pi/\lambda$, where $\lambda$ is the wavelength) shows that at wavenumbers smaller than 0.04 m$^{-1}$ (wavelengths longer than 160 m) and frequencies below 10 Hz, the cross-correlations no longer produce Rayleigh wave estimates (Supplementary Fig. 8). Our explanation is that at such wavenumbers, the equivalent wavelengths approach the length of the cable segment (220 m) and are no longer resolvable. Accordingly, our virtual Rayleigh waves are not sensitive to depths comparable to the glacier thickness.

## Data availability

Seismometer data of the 4D local glacier seismology network (https://doi.org/10.12686/sed/networks/4d/) are archived at the Swiss Seismological Service and can be accessed via its web interface http://arclink.ethz.ch/webinterface/. DAS data are archived at the ETH's Laboratory of Hydraulics, Hydrology and Glaciology (VAW), and access can be granted by the authors.

## Code availability

Our python implementation of matched-field processing is available at https://github.com/fablindner/glseis/blob/master/array_analysis.py. The NonLinLoc software can be downloaded at http://alomax.free.fr/nlloc/index.html.

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

## Acknowledgements

We thank M. Funk, R. Lörtscher, and the Swiss Seismological Service for help in the field and Edi Kissling for the discussions on 2D ray tracing. The fieldwork on Rhonegletscher and the salary of D.G. were financed via ETH Grant ETH-06 16-2. F.W., F.L., and M.C. were financed by the Swiss National Science Foundation via Grants PP00P2_157551 and PP00P2_183719. P.P. was funded through the ETH Grant "Distributed Acoustic Sensing" (Grant No. 1-001179-000).

## Author contributions

F.W. supported fieldwork, formulated most of the manuscript text, conducted MFP, and assisted in noise correlations. D.G. led seismometer and iDAS deployment, located the stick-slip event, and performed the cross-correlation search. F.L. analyzed the various stick-slip phases with M.K. who also participated in fieldwork. P.P. was responsible for DAS data acquisition in the field. M.C. performed the noise correlations and A.F. supervised the entire DAS data analysis and calculated the amplitude and phase response of the DAS system presented in the supplementary notes.

## Competing interests

The authors declare no competing interests.
