## [Peer Review File · Nature Communications]

Reviewers' comments:

Reviewer #1 (Remarks to the Author):

This paper presents a pioneering glacial seismology study using DAS fiber optic strain instrumentation. As such it is presently unique and undoubtedly points the way towards a new era of glacial seismology featuring tractable essentially massive multichannel data acquisition. I'd expect that this work would be highly cited in this context. The system performed well, and, as hoped, recorded a plethora of diverse signals from the glacier and its surroundings. The paper is very well written, contains substantially complete supplementary material, and reasonably comprehensive regarding the signals observed and some of the attendant analysis. It is an impressive proof of concept, but does not report a revolutionary new observation or insight. The paper is essentially publishable in this context as written (there are a few minor typos that could easily be corrected in production).

A criticism -- the response of the DAS system is impressive, but I would have appreciated a quantitative assessment of its noise level (e.g., what is the relative signal to noise and response of the fiber when compared as best as is possible to the co-sited velocimeters?) and an evaluation of spatial averaging effects of the relatively large (10 m) gauge length (I realize that this is a standard gauge length for DAS systems) when studying a wave field at this scale.

Reviewer #2 (Remarks to the Author):

This manuscript describes data recorded by seismometers and by a distributed acoustic sensing (DAS) on a glacier.

As shown in this work, DAS are easy to install and provide a large number of "sensors" with a wide frequency band.

But they are not as sensitive as seismometers and can be noisy.

The DAS allows for more accurate icequake locations than using a small network of 3 seismometers. It also allows identifying seismic phases (distinguishing between direct and refracted waves, detecting doubly reflected waveforms).

This study is interesting as it shows how DAS can be used for monitoring glaciers or gravitational instabilities.

It appears as a very promising and powerful tool.

The manuscript is also very well written and illustrated.

However, the work presented in this paper does not (yet) provide new information on glacier structure or on glacier dynamics.

This study could be improved by a more rigorous and complex analysis of data (more accurate estimation of seismic wave velocities, 3D model of bedrock topography).

- Velocity model.

It is not clear to me how VP and VS are chosen.

The values of VP and VS given on line 200 (VP=3800 m/s, VS=1900 m/s) are larger than the values

given on line 352 (VP=3750 m/s, VS=1875 m/s). Why?

Moreover, Dalban et al. (Journal of Glaciology 2016) measured much smaller values of VP=3330 +/- 80 m/s for the same glacier!

The seismic velocities used in this study (VP=3800 m/s, VS=1900 m/s, line 200) are significantly larger than values I obtained for another temperate glacier in the Alps. Using a network of 100 seismometers, I have measured VP=3590 m/s and VS=1830 m/s by minimizing time residuals (using P and S direct waves) for 38 basal icequakes on Argentière glacier (unpublished work). The icequake depths are in very good agreement with bedrock topography estimated from radar profiles.

I suspect that overestimating VP and VS may explain why the estimated icequake depth (165 m) is smaller than the glacier thickness (H=200 m, line 206).

Also, the seismograms for the explosion shown in Fig S5 suggest that the VP value is slightly over-estimated.

Why didn't you use the explosions in order to estimate VP and VS? Or used values estimated previously on the same glacier by Dalban et al (2016)?

Then the authors assume a 1D model in order to interpret seismic phases (Fig 4).

The bedrock topography is known from radar measurements. Although it is not very accurate (error of at least 10%, l362), using a 3D model of the ice-bed interface would be much more realistic than a simple 1D model.

Using a 3D model to identify seismic phases would provide more reliable arrival times and would be a good test to estimate the accuracy of the bedrock topography.

- Icequake locations

I am surprised by the very large location uncertainties obtained using the on-ice seismometers (l167 and Fig 4B).

Are all the red dots in Fig 4B consistent with the measured P and S arrival times within a picking error of 4 ms?

- Rockfalls

The authors show that the DAS can be used to detect and locate (source azimuth) a small rockfall.

The results are nice, but not surprising. Many previous studies have used beam-forming methods to locate rockfalls, and even follow the rockfall propagation (e.g., Lacroix and Helmstetter BSSA 2011).

- Discussion

Line 320 : "DAS technology could in the near future significantly lower detection thresholds and increase warning capabilities for destructive mass movements."

I agree that DAS could be used to detect many small events everywhere, but there are technical difficulties that should be discussed.

How to store and process automatically so much information?

While earthquakes can be detected and located automatically with a correct accuracy, detecting and locating mass movements is much more challenging. Indeed, seismic signals produced by mass movements are much more emergent than earthquake signals, and identifying (automatically) seismic phases is much more difficult.

Reviewer #3 (Remarks to the Author)

Walter and co-authors report results of a distributed acoustic sensing (DAS) survey on a glacier in the Swiss Alps. The authors show that DAS is able to recover signals from a variety of glacier-related processes such as a surface icequake and a stick-slip basal event, as well as noise from a rock fall occurring nearby. They are also able to recover signals from an active experiment using explosives drilled into the ice of the glacier at several hundreds of Hz. They successfully identify a variety of phases in the DAS recordings and show the added value of using the densely spaced DAS data to recover some phases that remain hidden in classic sensor recordings. To my knowledge, this is the first study focusing on using DAS to investigate glaciers. Results are very promising and could help advance the field of cryo/environmental seismology. The paper is well written and, in general, well presented.

I suggest publication after small changes and a few clarifications. In my opinion, the paper will benefit from a bit more description on the effects of the snow cover/ice-snow interface on the surface DAS signal and to what degree this affects the results. As I point out later in my revision, the waveforms from the DAS channels (deployed on the surface, on snow) are somewhat different to those recorded by the seismometers (deployed in boreholes in the ice). Expanding on the analysis of these differences would be helpful. I would also add a bit more discussion on the directionality of the DAS cable, and how this could affect locating surface icequakes and stick-slip events.

Comments on main text

1. Line 107. The authors clearly describe the layout of the experiment forming an equilateral triangle with 220 m side lengths. However, they do not give any reason why they decided on this geometry as opposed to other configurations such as one long profile, crossing transects, etc. Could you please elaborate on your choice of survey layout?

2. Lines 107 to 112. I suppose the seismometers are three-component sensors? Please specify as this becomes important when comparing the signals with the DAS recordings.

3. Lines 113-121. Given that this is the first reported DAS survey on a glacier, it might be useful to include some more comments on the deployment of the interrogator, such as: where was it located? How much power does it need in comparison with a seismometer (important for long-term deployments in this kind of environment, where power is not easily accessible)? Is it important to isolate the interrogator from temperature variations? This brief explanation would give the readers an idea of how much easier or challenging it is to perform DAS surveys in glaciers with respect to using classic sensors. This information could be added here or to the Discussion section.

4. Line 124. The authors make reference to “explosives at ca. 30 cm depth in the ice were set off”. It would be informative to briefly comment on aspects such as what the goal of the experiment was, how many explosions were set off, whether they were always set off at the same location or the source position was changed, what was the specific source used (i.e. size, expected frequency). Would it be possible to indicate the location of the explosions in Figure 1?

5. Line 137. Could you briefly describe the expected characteristics of a basal stick-slip event, here?

6. Lines 141-142. The authors state that they “compare the signal quality between DAS and seismometer”. Which component of the seismometer are you comparing with the DAS signal? Can you comment on the differences you would expect between the two signals given the directionality of DAS?

7. Lines 141-142. In relation to the previous comment – waveforms shown in Figure 2 for the seismometers and the DAS channel are quite different in some cases. What could be the origin of these differences? Coupling? Please elaborate on that.

8. Line 151. In relation to the previous question - the authors explain the high-frequency limit of the DAS data “ as the result of the damping snow layer”. Would this snow layer cause other effects on the signal recorded by DAS, besides damping the frequency content and amplitude of the signal? Are there any previous studies (using conventional sensors) reporting the effects of snow cover in seismic studies of glaciers?

9. Lines 151-152. The authors state that ‘Damping within the glacier ice is less severe as confirmed by the explosion signals’. This statement is somewhat confusing, could you elaborate a bit more? Signals from both the basal stick-slip event and the explosion need to go through glacier ice and thorough the snow layer to hit the DAS cable. How do you discern between damping caused by ice as opposed to that caused by the snow layer from looking at these signals?

10. Line 171-172. The authors state that the stick-slip event has a “source mechanism consistent with bed-parallel slip along the glacier flow line”. Have you calculated the source mechanism using the seismometers/DAS/both? Please clarify.

11. Line 200. Velocities are reported as “3800 and 1900 m/s for P and S-waves” here, but they are slightly different on the Materials and Methods section (lines 352 and 358-359). Although the values are very close, please be consistent in reporting the true values used in the modeling exercise.

12. Line 229. The authors suggest that the rock fall occurred at an “unstable moraine to the West of the glacier”. How far away is this moraine from the survey?

Comments on Materials and Methods

13. General comment on arrival time estimation and location. The velocity model used for these exercises only considers the ice of the glacier, but it neglects the ~ 3 m thick snow layer. Have you tried including that layer in your model? If not, do you expect it to have a significant effect on the arrival times estimated for the DAS recordings? Please comment on that.

14. Line 341. The authors report that they picked “as many P-wave arrivals as possible, mainly from the southern cable section”. Given the directionality of the DAS cable, P-waves are expected to be very weak on surface deployments. This could be a limitation for this type of exercise. Can you comment on the amplitudes of the P-waves observed in your records and how they compare with the seismometer records?

15. Line 411. The authors describe that they “spectrally whiten the signal”. Is this part of a pre-processing scheme? Please describe/mention any pre-processing applied to the data before applying MFP.

Comments on Figures

Figure 1, caption. Is the location of the stick-slip event on panel B based on DAS data analysis?

Figure 2, panel A. Based on the spectrogram shown in Figure S1, the surface icequake contains significant energy up to 50 Hz. Why do you choose an upper corner frequency of 30 Hz?

Figure 2, caption. “Note that the time axis between two seismograms of one panel were slightly shifted”. I guess you mean “for each panel”, or is it for one of the panels only? This phrase is confusing.

Figure 3 caption. The reference needs to be changed to number format.

Figure 4, panel A. This image can be confusing for readers who are not used to see DAS seismic profiles with co-located sections and symmetric signals. It would be useful to mark the location of channel 278 (mentioned in the caption) and the position of the triangle corners.

Figure 4, panel B. What's the color grid? It is not mentioned in the caption, and a colorscale is needed. Also, maybe a different color scheme would help since the colors are very similar to those of the different sections of the cable, which are very difficult to see.

Figure 4, panel D. The colors for the different phases are very similar and sometimes very difficult to see. It would also be useful to list the different phases in the legend in order of appearance from 0 time, when possible.

Figure 4, panels A and D. It would be better to display both panels in the same units of distance (channels or distance), or indicate in panel A which section of the profile is shown in panel D. As it is now, it is hard to relate the two figures.

Figure 4, caption. “ Green triangle and black lines” - did you mean colored lines?

Figure 5. It would be useful to add a colorscale for the normalized beam power values.

Comments on Supplementary material

Figures S1-S4. Are these the same recordings shown in Figure 2, or are they the same events but recorded on a different corner of the cable layout? In the caption of Figure 2, you state that you are showing events from the southern corner of the cable, whereas in the captions of Figures S1 to S4 you refer to the northwestern corner of the cable. If you are indeed showing different recordings, it would be more useful if the spectrograms shown here were those corresponding to the events in Figure 2. Please clarify.

Figure S3. It is slightly misleading as the y axis is different to that of all other spectrograms shown here. I recommend using the same y axis in all figures S1 to S4.

Figure S5. Figure label is missing.

Reviewer 1

This paper presents a pioneering glacial seismology study using DAS fiber optic strain instrumentation. As such it is presently unique and undoubtedly points the way towards a new era of glacial seismology featuring tractable essentially massive multichannel data acquisition. I'd expect that this work would be highly cited in this context. The system performed well, and, as hoped, recorded a plethora of diverse signals from the glacier and its surroundings. The paper is very well written, contains substantially complete supplementary material, and reasonably comprehensive regarding the signals observed and some of the attendant analysis. It is an impressive proof of concept, but does not report a revolutionary new observation or insight. The paper is essentially publishable in this context as written (there are a few minor typos that could easily be corrected in production).

A criticism -- the response of the DAS system is impressive, but I would have appreciated a quantitative assessment of its noise level (e.g., what is the relative signal to noise and response of the fiber when compared as best as is possible to the co-sited velocimeters?) and an evaluation spatial averaging effects of the relatively large (10 m) gauge length (I realize that this is a standard gauge length for DAS systems) when studying a wave field at this scale.

Noise and signal-to-noise: We investigated spatial changes in signal and noise strength for the stick-slip record shown in Figures 2-4. A new figure was added to the supplemental material and discussed in the main text (end of Section "RECORDED SIGNALS").

Spatial averaging: We now include a calculation on how the gauge length affects amplitude and phase response of the DAS system. For wavelengths longer than the gauge length as in our case, the responses do not affect arrival times. Note that this calculation led to new uncertainty estimates, which we updated in the manuscript accordingly.

Reviewer 2

This manuscript describes data recorded by seismometers and by a distributed acoustic sensing (DAS) on a glacier. As shown in this work, DAS are easy to install and provide a large number of "sensors" with a wide frequency band. But they are not as sensitive as seismometers and can be noisy. The DAS allows for more accurate icequake locations than using a small network of 3 seismometers. It also allows identifying seismic phases (distinguishing between direct and refracted waves, detecting doubly reflected waveforms).

This study is interesting at it shows how DAS can be used for monitoring glaciers or gravitational instabilities. It appears as a very promising and powerful tool. The manuscript is also very well written and illustrated. However, the work presented in this paper does not (yet) provide new information on glacier structure or on glacier dynamics. This study could be improved by a more rigorous and complex analysis of data (more accurate estimation of seismic wave velocities, 3D model of bedrock topography).

- Velocity model.

It is not clear to me how VP and VS are chosen. The values of VP and VS given on line 200 (VP=3800 m/s, VS=1900 m/s) are larger than the values given on line 352 (VP=3750 m/s, VS=1875 m/s). Why? Moreover, Dalban et al. (Journal of Glaciology 2016) measured much smaller values of VP=3330 +/- 80 m/s for the same glacier! The seismic velocities used in this study (VP=3800 m/s, VS=1900 m/s, line 200) are significantly larger than values I obtained for another temperate glacier in the Alps. Using a network of 100 seismometers, I have measured VP=3590 m/s and VS=1830 m/s by minimizing time residuals (using P and S direct waves) for 38 basal icequakes on Argentière glacier (unpublished work). The icequake depths are in very good agreement with bedrock topography estimated from radar profiles.

We reran the location inversion with the same seismic velocities as for ray tracing. As the ray tracing depends on the locations, the results changed slightly. The individual phases are still fit satisfactorily, but we had to significantly increase the S-wave velocity of the bedrock to 3200 m/s. Since near-surface granite rock is known to have variable seismic velocities depending on weathering and layer orientation, e.g., this is not an unreasonable value (e.g. Živor et al., 2011, in *Acta Geodynamica and Geomaterialia*). We did remove the reference to a previous study of moment tensor inversions. Furthermore, we also added a statement and references to explain slower englacial seismic velocities in previous studies. This includes the study by Dalban et al. (2016) whose low seismic velocities describe wave propagation in the top ice layer and can be explained with surface crevassing known to lower seismic velocities. In general, we agree that our seismic velocities are on the high end of what we expect, but they nevertheless agree with previously published results.

I suspect that overestimating VP and VS may explain why the estimated icequake depth (165 m) is smaller than the glacier thickness (H=200 m, line 206). Also, the seismograms for the explosion shown in Fig S5 suggest that the VP value is slightly over-estimated. Why didn't you use the explosions in order to estimate VP and VS? Or used values estimated previously on the same glacier by Dalban et al (2016)?

Unfortunately, reducing the seismic velocities would lead to even later reflection arrivals with respect to the direct waves. To maintain the relative arrival times the thickness estimate would be even smaller. It is important to note that 165 m is not the icequake depth but rather the relative hypocenter location in a tilted reference system. As far as the explosions, please refer to the previous comment: Similar to shallow seismic sources used in Dalban et al. (2016), our explosion records on near-surface seismometers and the DAS cable sample seismic waves, which travel primarily through the low-velocity surface crevasse zone. We therefore did not use the explosions to determine seismic velocities for the location inversion.

Then the authors assume a 1D model in order to interpret seismic phases (Fig 4). The bedrock topography is known from radar measurements. Although it is not very accurate (error of at least 10%, l362), using a 3D model of the ice-bed interface would be much more realistic than a simple 1D model. Using a 3D model to identify seismic phases would provide more reliable arrival times and would be a good test to estimate the accuracy of the bedrock topography.

Although a 3D model is more realistic than a 1D model we cannot guarantee that in our case it represents the glacier thickness better, because of the uncertainties of available radar profiles. In this case, the use of a complex velocity model would be inappropriate, as 3D velocity models would introduce similar bias in location as the homogeneous one. Since there is a trade-off between unknown seismic velocities and glacier thickness we find it difficult to argue for a “best model” and therefore prefer not to move to the 3D case, which is substantially more difficult to implement. In addition, to make the 3D model complete, one would need to model the snow layer as well (see response to Reviewer 3 below). The unknown variations in snow depth over the DAS array and the highly variable and largely unconstrained seismic velocities in snow would make it more a guess than a well-founded 3D model.

- Icequake locations

I am surprised by the very large location uncertainties obtained using the on-ice seismometers (I167 and Fig 4B). Are all the red dots in Fig 4B consistent with the measured P and S arrival times within a picking error of 4 ms?

From our experience, such large side lobes emerge when few stations are available for arrival time measurements. Accordingly, they result from the nonlinear nature of the probabilistic inversion problem given uncertainties in arrival time measurements and the velocity model.

- Rockfalls

The authors show that the DAS can be used to detect and locate (source azimuth) a small rockfall. The results are nice, but not surprising. Many previous studies have used beam-forming methods to locate rockfalls, and even follow the rockfall propagation (e.g., Lacroix and Helmstetter BSSA 2011).

We now point out and cite this study.

- Discussion

Line 320 : "DAS technology could in the near future significantly lower detection thresholds and increase warning capabilities for destructive mass movements." I agree that DAS could be used to detect many small events everywhere, but there are technical difficulties that should be discussed. How to store and process automatically so much information? While earthquakes can be detected and located automatically with a correct accuracy, detecting and locating mass movements is much more challenging. Indeed, seismic signals produced by mass movements are much more emergent than earthquake signals, and identifying (automatically) seismic phases is much more difficult.

We included a new paragraph and supplemental figure (Fig. S10) that offers some perspectives on DAS data processing. These include machine learning, array techniques and signal stacking, whose efficiency is demonstrated in the new supplemental figure.

Reviewer 3

Revision of manuscript NCOMMS-19-33791: "Distributed Acoustic Sensing of Microseismic Sources and Wave Propagation in Glaciated Terrain" by Walter et al. Walter and co-authors report results of a distributed acoustic sensing (DAS) survey on a glacier in the Swiss Alps. The authors show that DAS is able to recover signals from a variety of glacier-related processes such as a surface icequake and a stick-slip basal event, as well as noise from a rock fall occurring nearby. They are also able to recover signals from an active experiment using explosives drilled into the ice of the glacier at several hundreds of Hz. They successfully identify a variety of phases in the DAS recordings and show the added value of using the densely spaced DAS data to recover some phases that remain hidden in classic sensor recordings. To my knowledge, this is the first study focusing on using DAS to investigate glaciers. Results are very promising and could help advance the field of cryo/environmental seismology. The paper is well written and, in general, well presented.

I suggest publication after small changes and a few clarifications. In my opinion, the paper will benefit from a bit more description on the effects of the snow cover/ice-snow interface on the surface DAS signal and to what degree this affects the results. As I point out later in my revision, the waveforms from the DAS channels (deployed on the surface, on snow) are somewhat different to those recorded by the seismometers (deployed in boreholes in the ice). Expanding on the analysis of these differences would be helpful. I would also add a bit more discussion on the directionality of the DAS cable, and how this could affect locating surface icequakes and stick-slip events.

Comments on main text

1. Line 107. The authors clearly describe the layout of the experiment forming an equilateral triangle with 220 m side lengths. However, they do not give any reason why they decided on this geometry as opposed to other configurations such as one long profile, crossing transects, etc. Could you please elaborate on your choice of survey layout?

We chose this study site, because previous monitoring had confirmed the existence of basal seismicity in this region. Moreover, we used the triangular layout to compare seismometer and DAS records and the seismometers were part of a multi annual array on the glacier. This is now described in the text.

2. Lines 107 to 112. I suppose the seismometers are three-component sensors? Please specify as this becomes important when comparing the signals with the DAS recordings. 3. Lines 113-121. Given that this is the first reported DAS survey on a glacier, it might be useful to include some more comments on the deployment of the interrogator, such as: where was it located? How much power does it need in comparison with a seismometer (important for long-term deployments in this kind of environment, where power is not easily accessible)? Is it important to isolate the interrogator from temperature

variations? This brief explanation would give the readers an idea of how much easier or challenging it is to perform DAS surveys in glaciers with respect to using classic sensors. This information could be added here or to the Discussion section.

We now specify that the on-ice and on-rock seismometers are three-component sensors. Unfortunately, the interrogator manufacturer only granted us permissions to offer rough technical specifications on power assumptions. Therefore, we only included a general but hopefully helpful statement on the power supply.

4. Line 124. The authors make reference to “explosives at ca. 30 cm depth in the ice were set off”. It would be informative to briefly comment on aspects such as what the goal of the experiment was, how many explosions were set off, whether they were always set off at the same location or the source position was changed, what was the specific source used (i.e. size, expected frequency). Would it be possible to indicate the location of the explosions in Figure 1?

The active experiment was an exploratory exercise. We now offer specific information in the supplemental material (Fig. S5 caption).

5. Line 137. Could you briefly describe the expected characteristics of a basal stick-slip event, here?

These characteristics are specified in the following paragraph, which we now point out.

6. Lines 141-142. The authors state that they “compare the signal quality between DAS and seismometer”. Which component of the seismometer are you comparing with the DAS signal? Can you comment on the differences you would expect between the two signals given the directionality of DAS?

Please see reply to Comment 8.

7. Lines 141-142. In relation to the previous comment – waveforms shown in Figure 2 for the seismometers and the DAS channel are quite different in some cases. What could be the origin of these differences? Coupling? Please elaborate on that.

Please see reply to Comment 8.

8. Line 151. In relation to the previous question - the authors explain the high-frequency limit of the DAS data “ as the result of the damping snow layer”. Would this snow layer cause other effects on the signal recorded by DAS, besides damping the frequency content and amplitude of the signal? Are there any previous studies (using conventional sensors) reporting the effects of snow cover in seismic studies of glaciers?

Although many cryoseismic studies have been performed on the Antarctic Ice Sheet on top of a thick snow/firn layer, we are not aware of studies systematically analyzing the snow effect. We elaborate on snow damping (see next comment) and now include another panel to supplemental Figure S5.

In general, at frequencies below 100 Hz, wavelengths of 10's of meters will not be sensitive to the snow layer. The reason why seismometer and DAS records differ is more likely related to directionality: For the seismic records we show vertical ground velocity, whereas for the DAS cable we show horizontal strain along the cable axis. We can test the expected proportionality between ground velocity and DAS measurement mentioned in the manuscript for our stick-slip event from Figures 2-4: the two time series agree reasonably well as shown in the figure below when normalized and filtered between 30 and 70 Hz. (DAS in red and seismometer in black). The shown horizontal seismometer record is unknown (borehole sensors) but the good agreement with the DAS record means that it is approximately aligned with the cable axis. A systematic study requires an independent seismometer orientation, e.g. with the help of P-phases from icequake records. This is subject to ongoing work and we plan to publish this subsequently.

9. Lines 151-152. The authors state that ‘Damping within the glacier ice is less severe as confirmed by the explosion signals’. This statement is somewhat confusing, could you elaborate a bit more? Signals from both the basal stick-slip event and the explosion need to go through glacier ice and thorough the snow layer to hit the DAS cable. How do you discern between damping caused by ice as opposed to that caused by the snow layer from looking at these signals?

Apparently, our argumentation was unclear, so thank you for pointing this out. We reworded this section and refer to explosion reflections, which are visible on the seismometers but not on the DAS system (new panel to Fig. S5).

10. Line 171-172. The authors state that the stick-slip event has a “source mechanism consistent with bed-parallel slip along the glacier flow line”. Have you calculated the source mechanism using the seismometers/DAS/both? Please clarify.

We used a slip, which is bed-parallel and in the direction of glacier flow. This is consistent with compressive direct P-waves at all on-ice seismometers. This is now specified.

11. Line 200. Velocities are reported as “3800 and 1900 m/s for P and S-waves” here, but they are slightly different on the Materials and Methods section (lines 352 and 358-359). Although the values are very close, please be consistent in reporting the true values used in the modeling exercise.

This was changed and consistent values are now used (see response to Reviewer 2).

12. Line 229. The authors suggest that the rock fall occurred at an “unstable moraine to the West of the glacier”. How far away is this moraine from the survey?

We now specify an approximate distance of 1 km.

Comments on Materials and Methods

13. General comment on arrival time estimation and location. The velocity model used for these exercises only considers the ice of the glacier, but it neglects the ~ 3 m thick snow layer. Have you tried including that layer in your model? If not, do you expect it to have a significant effect on the arrival times estimated for the DAS recordings? Please comment on that.

It is difficult to include the snow layer in the model, because a simple 1D layered model is not applicable to the three dimensional shape of the fiber optics layout (see figure below). This requires a 3D model, which carries additional uncertainties associated with snow depth variations. The effect of the 3 m snow layer on hypocenter location is estimated as follows. We calculate the travel time difference between a direct wave through ice, only, and a wave through ice and a snow layer, representing the difference between the reality and our model:

$$\Delta t \leq \frac{\sqrt{d^2+h_i^2}}{v_i} + \frac{h_s}{v_s} - \frac{\sqrt{d^2+h^2}}{v_i}, \quad (1)$$

where d is the epicentral distance between source and sensor, h is the vertical distance between hypocenter and sensor, h_s is the snow layer thickness, h_i is the ice thickness, v_i is the wave velocity in ice, v_s is the velocity in snow and $h = h_i + h_s$. The first term represents the travel time of a wave through the ice and the second term is the time of vertical propagation through the snow layer. The third term is our model’s travel time approximation assuming that the snow layer is filled with ice. The near-vertical travel direction within the snow layer is justified since the wave velocities in snow are slower by a factor of 5-10 (Fermat’s principle and Snell’s law). With $h = 190$ m, $h_s = 3$ m we find travel time differences between 2.5-4 ms for P-waves and 8-10 ms for S-waves translating to a hypocenter uncertainty of up to 15 m.

There exists a high uncertainty of >100% for elastic wave velocities in snow (Capelli et al., 2016, in *Cold Regions Science and Technology*). Furthermore, the variation in snow layer thickness is >10%. On top of

that the bedrock uncertainty is also assumed to be larger than >10%. All this results in a hypocenter uncertainty estimate, that is comparable to the uncertainties from our 1D layer model location.

14. Line 341. The authors report that they picked “as many P-wave arrivals as possible, mainly from the southern cable section”. Given the directionality of the DAS cable, P-waves are expected to be very weak on surface deployments. This could be a limitation for this type of exercise. Can you comment on the amplitudes of the P-waves observed in your records and how they compare with the seismometer records?

The relative amplitudes of P- and S-waves result from the double couple radiation pattern and the angle between seismic wave polarization and cable axis. We now specify this in the text and in the caption of Figure 4.

15. Line 411. The authors describe that they “spectrally whiten the signal”. Is this part of a preprocessing scheme? Please describe/mention any pre-processing applied to the data before applying MFP.

Keeping only the phase information in the cross spectral density matrix amounts to spectral whitening. We rewrote this part of the text to avoid the impression that we pre-process the data.

Comments on Figures

Figure 1, caption. Is the location of the stick-slip event on panel B based on DAS data analysis?

Yes, this is now specified.

Figure 2, panel A. Based on the spectrogram shown in Figure S1, the surface icequake contains significant energy up to 50 Hz. Why do you choose an upper corner frequency of 30 Hz?

We decided on the 30 Hz corner, because our previous study on Alpine icequakes (Walter et al., 2009, BSSA) showed that in the 5-30 Hz the retrograde Rayleigh wave ellipse is most apparent.

Figure 2, caption. "Note that the time axis between two seismograms of one panel were slightly shifted". I guess you mean "for each panel", or is it for one of the panels only? This phrase is confusing.

We changed the figure and now only shift the time axes of Panels A-C. This is now specified.

Figure 3 caption. The reference needs to be changed to number format.

Done.

Figure 4, panel A. This image can be confusing for readers who are not used to see DAS seismic profiles with co-located sections and symmetric signals. It would be useful to mark the location of channel 278 (mentioned in the caption) and the position of the triangle corners.

We marked the location of Channel 278 with a green arrow. Moreover, we changed the channel labels to the convention used in other parts of the manuscript, e.g. "D276", which should make it easier for the reader to identify channel locations (especially given the new supplemental figure S5).

Figure 4, panel B. What's the color grid? It is not mentioned in the caption, and a colorscale is needed. Also, maybe a different color scheme would help since the colors are very similar to those of the different sections of the cable, which are very difficult to see. Figure 4, panel D. The colors for the different phases are very similar and sometimes very difficult to see. It would also be useful to list the different phases in the legend in order of appearance from 0 time, when possible.

Color grid: this corresponds to elevation and is only for illustration purposes. We now specify this. A colorbar would be redundant, because color is already defined by the z-axis. We did change the view angle and included the location of the second stick-slip event shown in the stacking exercise (Figure

S10). Furthermore, we changed color of the phase arrivals and legend order and we removed some theoretical arrival times to make the panel more readable.

Figure 4, panels A and D. It would be better to display both panels in the same units of distance (channels or distance), or indicate in panel A which section of the profile is shown in panel D. As it is now, it is hard to relate the two figures.

All traces of Panel A are shown in Panel D, just in a different illustration. We now mark the cable end in Panel A. Also, the colored triangle sides in Panel B are indicated more clearly now and should help better identify channel locations. Furthermore, we changed the y-labels in Panel A to trace names that are referred to in other parts of the text.

Figure 4, caption. "Green triangle and black lines" - did you mean colored lines?

This part of the caption belonged to an old figure version (thanks for catching this). We removed the station triangles and now only show the cable triangle.

Figure 5. It would be useful to add a colorscale for the normalized beam power values.

The color scale is exactly defined by the y-axis in Panel D.

Comments on Supplementary material

Figures S1-S4. Are these the same recordings shown in Figure 2, or are they the same events but recorded on a different corner of the cable layout? In the caption of Figure 2, you state that you are showing events from the southern corner of the cable, whereas in the captions of Figures S1 to S4 you refer to the northwestern corner of the cable. If you are indeed showing different recordings, it would be more useful if the spectrograms shown here were those corresponding to the events in Figure 2. Please clarify.

We changed Figures 2 and 3 as well as Figures S1-S3 and now show DAS records from channel D620 (southern triangle corner). The only exception is the rock fall signal, which is shown at the western corner where it has a higher signal-to-noise ratio. Thus, the supplemental figures now show the same signals as in Figure 2.

Figure S3. It is slightly misleading as the y axis is different to that of all other spectrograms shown here. I recommend using the same y axis in all figures S1 to S4.

We were aware that the different y-scales in the spectrograms Figures S1-S4 may be misleading. At the same time, when using the same y-scales, some important differences in spectral signature are less

obvious. For this reason we decided to stick with different y-scales, which we now point out in Figures S3 and S4.

Figure S5. Figure label is missing.

Done.

Reviewers' comments:

Reviewer #1 (Remarks to the Author):

I have re-read the (revised) manuscript. This is a solid description and early analysis of a pioneering data set collected in an alpine environment with a DAS/seismic system. As aptly noted in the paper, this technological breakthrough has numerous implications for improving knowledge and practical understanding of glaciological and alpine seismicity. The paper was clearly improved in a number of details following review. As an early exemplar of DAS data and analysis recorded on a glacier, I expect that this paper will be influential in accelerating already rapidly growing awareness and greater interest in applying this technology and its unique data acquisition capabilities in cryoseismology. More generally DAS systems are clearly engendering a data acquisition breakthrough in seismology and broader associated Earth science that I expect to be one of the principal advancements for the coming decade.

Reviewer #2 (Remarks to the Author):

The authors have responded to most of my questions, except on the velocity model. The authors still use rather high seismic velocities in the ice, and do not explain in the text how these values are chosen. They say that these values are "typical", but do not provide any reference. In the rebuttal letter, the authors explained that seismic wave velocities were adjusted to match both direct and reflected arrival times for the stick-slip signal. For locating the basal icequake, I agree with the choice of a homogeneous velocity model, since only direct waves are used. However, when modeling reflected waves, I don't think it is realistic to use a 1D model (flat ice layer above the bedrock). The arrival time of the reflected wave depends on the bedrock slope, which is 25° according to the 3D model based on radar data, but is assumed to be horizontal in the 1D model. I also don't understand why you don't use the explosions to estimate seismic wave velocities: you show a nice reflected wave in Figure S5B. Why can't you use it to invert for the glacier thickness and/or the P wave velocity? Of course, using a 1D model is much simpler, but it's a pity to have so nice data and not using a more realistic model to interpret it. This study is promising ... but I am a bit disappointed that this experience is not (yet) used to provide new information on the glacier structure or dynamics.

Reviewer #3 (Remarks to the Author):

The revised version of the manuscript is an improvement with respect to the original version. The authors have carefully addressed all of my concerns responding to my questions and also including further details in the manuscript text, modifying figures and including additional figures in the Supplementary material. I appreciate the effort of the authors on carrying out additional analysis and even adding an extra note with further details on some of the more theoretical aspects of the system. I believe the paper is a nice proof of concept and is now of greater use to other scientists wanting to reproduce similar DAS experiments in glaciated regions. In my opinion, the paper is now publishable in its current form (except for a couple of typos).

Reviewer 2:

The authors still use rather high seismic velocities in the ice, and do not explain in the text how these values are chosen. They say that these values are "typical", but do not provide any reference. In the rebuttal letter, the authors explained that seismic wave velocities were adjusted to match both direct and reflected arrival times for the stick-slip signal.

For locating the basal icequake, I agree with the choice of a homogeneous velocity model, since only direct waves are used. However, when modeling reflected waves, I don't think it is realistic to use a 1D model (flat ice layer above the bedrock). The arrival time of the reflected wave depends on the bedrock slope, which is 25° according to the 3D model based on radar data, but is assumed to be horizontal in the 1D model.

I also don't understand why you don't use the explosions to estimate seismic wave velocities: you show a nice reflected wave in Figure S5B. Why can't you use it to invert for the glacier thickness and/or the P wave velocity?

Of course, using a 1D model is much simpler, but it's a pity to have so nice data and not using a more realistic model to interpret it.

This study is promising ... but I am a bit disappointed that this experience is not (yet) used to provide new information on the glacier structure or dynamics.

As encouraged by Reviewer 2, we explored the use of a 2D seismic velocity model in order to calculate arrival times of direct, multiply reflected and refracted waves. In order to incorporate the results of this exercise we completely rewrote the methods sections, included an additional supplementary figure, changed Figure S5 and made several adjustments to the text.

The 2D model was picked along a longitudinal glacier cross-section connecting the basal icequake epicenter with the southernmost corner of the fiber optic cable. To mitigate the effect of interpolation between radar profiles (spaced by hundreds of meters), which may introduce spurious steps or even local minima in the bed topography, we apply a 30 m-wide moving average filter to the bed profile. A figure showing the cross-sectional profile is now included in the supplemental material (Figure S7).

The smoothed longitudinal bed profile shows 2D features, such as inclination and overdeepening near the network and event epicenter. The effects of these geometric features are somewhat limited as the surface is also inclined, which we compensate by rotating the along-cross-section coordinates counterclockwise. Moreover, we calculate other parallel longitudinal profiles, which intersect the cable and find significant variations in bed height (up to 60 m) and surface height (up to 4.5 m) among these parallel profiles suggesting additional 3D effects not captured in our cross section. Nevertheless, we work with the central 2D cross section for our raytracing calculations.

Using a 2D ray-shooting algorithm we calculate and plot the arrival times of doubly reflected body waves and refracted S-waves. As with the 1D case we find the high P- and S-velocities are needed to match the steep slopes of the indirect arrivals, even when we use a 24 m thinner bed than what the smoothed longitudinal cross section gives. Shifting the bedrock height, hypocenter and/or origin time (all of which have uncertainties associated with location and radar profiles) does not produce a steep enough phase moveout when using lower body wave velocities. In fact, this effect is more pronounced in the 2D case as the overdeepening further delays indirect arrival times. This leads to slightly worse arrival time matching shown in Figure 4D. We can reduce (though not eliminate) this effect by

elevating the reflecting bed region by an additional 3 m. Such a local thinning is somewhat arbitrary but may be related to reflections that occur off-axis with respect to the longitudinal cross-section (bed gradients are steepest in the transverse direction).

In the revised manuscript version we discuss details of our raytracing strategy, which produces these relatively high seismic velocities (we should have done this for the 1D case, too). We acknowledge that the velocities are higher than what we expected.

The high seismic velocities also brought out in our 2D raytracing prompted us to modify the discussion: First, we specify a more modest scope for the raytracing arguing that it helps interpret different phases, whereas a tight quantitative match is not achieved. Second, we specify the major weakness of the cable set-up and source-location geometry: for a joint hypocenter-velocity inversion, our azimuthal gap exceeding 270 degrees is too high (Haslinger et al., 1999, now cited). Third, we explain that our raytracing hinges on matching the doubly reflected S-wave, even though it has a weak amplitude and is only visible on a subset of stations. Abandoning the requirement to match the double S-reflection would allow for lower seismic velocities. A possible solution to this trade-off could be a different interpretation of the double S-reflection. It is possible that this phase does not only contain waves, which travel within the ice, but also critically refracted waves. Accordingly, the wave may first travel as a critically refracted phase along the ice-bed interface and then enter the ice medium, upon which it undergoes the two reflections. Since the bed velocity is substantially faster, this would decrease the arrival time and a lower S-velocity within the ice would be permissible. What speaks for this scenario is that the phase moveout shown in Figure 4A and D seems more similar (curvature in Panel A “more parallel”) to the moveout of the refracted S-wave than the direct S-wave. In order to further investigate this possibility, we feel that raytracing is inadequate as wave amplitudes and phases should be used, too, which would require full waveform modeling. We nevertheless discuss this matter in the methods section. A final point is the possibility of 3D effects, which are not captured in our longitudinal bed profile. Given the uncertainties in source location and bed elevation we feel that going beyond the 2D approximation is not reasonable.

Best,
Fabian Walter.

Reviewers' comments:

Reviewer #1 (Remarks to the Author)

I have re-read the (revised) manuscript. This is a solid description and early analysis of a pioneering data set collected in an alpine environment with a DAS/seismic system. As aptly noted in the paper, this technological breakthrough has numerous implications for improving knowledge and practical understanding of glaciological and alpine seismicity. The paper was clearly improved in a number of details following review. As an early exemplar of DAS data and analysis recorded on a glacier, I expect that this paper will be influential in accelerating already rapidly growing awareness and greater interest in applying this technology and its unique data acquisition capabilities in cryoseismology. More generally DAS systems are clearly engendering a data acquisition breakthrough in seismology and broader associated Earth science that I expect to be one of the principal advancements for the coming decade.

Reviewer #2 (Remarks to the Author)

The authors have responded to most of my questions, except on the velocity model. The authors still use rather high seismic velocities in the ice, and do not explain in the text how these values are chosen. They say that these values are "typical", but do not provide any reference. In the rebuttal letter, the authors explained that seismic wave velocities were adjusted to match both direct and reflected arrival times for the stick-slip signal. For locating the basal icequake, I agree with the choice of a homogeneous velocity model, since only direct waves are used. However, when modeling reflected waves, I don't think it is realistic to use a 1D model (flat ice layer above the bedrock). The arrival time of the reflected wave depends on the bedrock slope, which is 25° according to the 3D model based on radar data, but is assumed to be horizontal in the 1D model. I also don't understand why you don't use the explosions to estimate seismic wave velocities: you show a nice reflected wave in Figure S5B. Why can't you use it to invert for the glacier thickness and/or the P wave velocity? Of course, using a 1D model is much simpler, but it's a pity to have so nice data and not using a more realistic model to interpret it. This study is promising ... but I am a bit disappointed that this experience is not (yet) used to provide new information on the glacier structure or dynamics.

Reviewer #3 (Remarks to the Author)

The revised version of the manuscript is an improvement with respect to the original version. The authors have carefully addressed all of my concerns responding to my questions and also including further details in the manuscript text, modifying figures and including additional figures in the Supplementary material. I appreciate the effort of the authors on carrying out additional analysis and even adding an extra note with further details on some of the more theoretical aspects of the system. I believe the paper is a nice proof of concept and is now of greater use to other scientists wanting to reproduce similar DAS experiments in glaciated regions. In my opinion, the paper is now publishable in its current form (except for a couple of typos).

We thank Reviewer 2 (Dr. Agnès Helmstetter) for the 3rd round of reviews. The reviewer requires going beyond the 2D model for raytracing which we employed in the last round of revisions. The main reason seems to be that she does not agree with our P- and S-wave velocities of ice, which to her mind are too high, and the approach we take to arrive at these velocities. Whereas we acknowledge that our study is not ideal for determining englacial seismic velocities, we do not see a point in adding more complexity to our calculations. Neither the source-station geometry, nor the bedrock profile accuracy are adequate for reliable calculations of englacial velocities. 3D raytracing may be a more fancy technique than our 2D approach, but this in itself will not increase the credibility of our results. In general, one can always add model parameters and complexity to a poorly constrained inversion problem, but this does not improve the scientific quality of the investigation. Moreover, we doubt that existing numerical solvers for 3D ray tracing can be applied in the presence of strong topographic variations of velocity contrasts found at the bed of an Alpine glacier. Since the dimension of the raytracing geometry does not affect any of our study's main messages, we see no reason for moving to the 3D case. Below we detail our arguments as well as responses to the reviewer's minor remarks.

Fabian Walter on behalf of the authors.

Our arguments fall into three categories: (1) There is no method or code available for 3D raytracing in media with strong topographic variations. (2) Though our velocities are slightly higher than in some previous studies, they are not higher than what is expected from theory and current, independent research on Rhonegletscher. (3) While some of the observed arrivals may be interpreted differently, the underlying model would need to be considerably more complex, thus also making it much less likely than the simple explanation that we currently favor. Please find more details below:

(1) In her review, Dr. Helmstetter herself admits that she is actually not aware of a method or code for 3D raytracing in the presence of topographic variations (surface and bedrock, in our case). Also, to the best of our knowledge, there is currently no option. To be sure, we contacted Dr. Nick Rawlinson at the University of Cambridge, who is one of the top experts on ray tracing in complex media. Dr. Rawlinson confirmed that even the most advanced fast-marching solvers of the eikonal equation would only allow us to model first-arriving P- and S-waves in a medium such as Rhonegletscher. Thus, the multiply reflected phases that we are discussing here cannot be simulated. This means that Dr. Helmstetter is indeed asking us to perform calculations that are currently not at all possible.

(2) In general, we are not aware of an accepted range of englacial seismic velocities for temperate glacier ice, at least not in the published literature. The range of published seismic velocities in ice is large. While the reviewer states that she has determined lower seismic velocities on an Alpine glacier and the first author of our study also used slower values for seismic signals on Gornergletscher (Switzerland), there exist observational and theoretical studies arriving at similar or even higher velocities than our study. An ongoing PhD project by S. Hellmann (ETH Zürich) measuring velocities on Rhonegletscher in boreholes shows preferred crystal orientation with S-velocities above 1800 m/s. This can be seen in the shot gather shown in Figure 1, where the dotted line fits the S-wave as a function of source-station distance. The

colors show some variation in S-velocity with this distance, but all values are above 1800 m/s reaching or exceeding 1900 m/s (our value). While these are unpublished data, in the revised manuscript we now discuss in the Materials and Methods section the range of seismic velocities in Alpine glacier ice arguing that our velocities are neither uncommon nor the highest expected values (note that we moved other text portions dealing with seismic velocities into the Materials and Methods section, too).

Figure 1: Cross-hole shot gather on Rhonegletscher. S-arrival time fit (blue dotted line) indicates variations in S-velocities (color coded), which reach or exceed the value found in our study (1900 m/s). Unpublished data from S. Hellmann, ETH Zürich.

In summary, we do not agree that our values for P- and S-velocities are unreasonable or even high compared to other studies. Looking for a poorly constrained 3D seismic velocity model, which suggests such high values is not justified. The velocities found by the reviewer's study mean to us that there may be important differences in ice properties between different glaciers, and we see no point in arguing these differences away.

(3) Apart from our arguments on seismic velocities, we want to emphasize our previous argument that a 3D raytracing velocity model - if it were technically possible - would not provide an improvement with respect to the 2D or even 1D approximation. As stated in our last response to the reviewer's comments, the basal topography which we rely on is based on sparse radar transects (see Figure 2 below: blue for

2008 helicopter-borne radar transects, green for 2008 ground-based radar transects, red for 2003 ground-based radar transects). Together with a typical uncertainty of at least 10% of radar-derived ice thicknesses, this means that the bed profile is simply not known accurately enough to drive 2D or 3D seismic velocity ray tracing. Moreover, as shown for the 2D approximation, the multiple S-reflection modelling will not improve with the 3D model as a result of the bed over-deepening. Therefore, we will have to shift the velocity model to achieve a good fit, and even more so if we use the slower seismic velocity suggested by the reviewer. The bottom line is that we will have to abandon the simple multiple reflection explanation for this phase. In this case, the original 1D velocity model would have explained the other phases adequately and including 2D topography would not have been necessary in the first place. Essentially, a more complicated velocity model will leave more room for explaining arrival times with complicated seismic phases, but also erodes the credibility of our explanations.

Figure 2: Radar transects. Blue for 2008 helicopter-borne radar transects, green for 2008 ground-based radar transects, red for 2003 ground-based radar transects.

At this point, we have the impression that the review process may no longer serve its purpose. We are asked to explore a research direction that is technically impossible and for which our data are not suited. Moreover, the remaining point of contention on small details in seismic velocities has no impact at all on the central message of our paper that distributed acoustic sensing provides unprecedented data for cryoseismological applications.

MINOR POINTS

Fig 4D still shows arrival times estimated using a 1D model.

For consistency, you should use the same velocity model (2 or 3D...) as used elsewhere (Fig S7).

This is a typo: we had not modified the caption of this figure although the 2D model was used for ray tracing. This is now corrected.

Fig S5. The green line shows the arrival time of the direct P wave for $V_P=3800$ m/s.

For positive distances, it seems to under-estimate arrival times, suggesting a lower P wave velocity (at least for this shallow source...).

But the fit for negative distances seems better. Do you understand why?

Could you use this data and the reflected P wave shown in Fig S5B to estimate V_P ?

We had noticed this, too. Our favorite explanation is that there exist lateral velocity variations resulting from differences in fracture density, although it should be noted that this regions contains few crevasses. We now offer this explanation in the caption.

Using this shot gather to estimate seismic velocities again requires a reflection depth, i.e. a 3D seismic velocity model, which we do not know accurately enough. Moreover, the reflected and direct P-waves do not travel at the same velocities as near-surface fractures are known to substantially slow down seismic waves (e.g. Deichmann et al., 2000, cited in the main text).

Fig S7. "30-point": I guess you mean 30 m?

No, we actually use a 30-point smoothing kernel to isolate bed inclination and curvature of the bed topography (Figure 3). This corresponds to a 342 m kernel, which we now specify. We had incorrectly stated a 30 m wide kernel in the Materials and Methods section, so thank you for catching this. Without our smoothing, the interpolated radar profiles yield topography steps over length scales near the seismic wavelengths of interest. These steps are likely unphysical and our 2D ray tracer cannot handle them numerically (we are no longer confident in the ray geometry).

Figure 3: 2D surface / bed cross-section with (black) and without (blue) smoothing. Star indicates stick-slip hypocenter.

L458 "critically reflected phase": do you mean "critically refracted phase"?

Yes, this is now corrected.

FIGURES

In Figure 4D we no longer show the moveout calculations of the doubly reflected S-waves for offsets below 270 m. At such small distances the cable channels locate more than 50 m from the cross-section axis used to calculate the 2D velocity model. The northern triangle side is an exception, with closer channels to the cross-section axis, but the S-reflection is weak there.

We noticed that we did not use the exact hypocenter location (red star in Figure S7) to calculate arrival times of the 2D ray tracing. Even though we did not notice a visible change in the resulting figures, we regenerated Figures 4D and S7 with the exact location.